# Sparse tree-based Initialization for Neural Networks

**Patrick Lutz**[1,*]  **Ludovic Arnould**[2]  **Claire Boyer**[2]  **Erwan Scornet**[3]
[1]Boston University  [2]LPSM, Sorbonne University  [3]CMAP, École Polytechnique

## Abstract

Dedicated neural network (NN) architectures have been designed to handle specific data types (such as CNN for images or RNN for text), which ranks them among state-of-the-art methods for dealing with these data. Unfortunately, no architecture has been found for dealing with tabular data yet, for which tree ensemble methods (tree boosting, random forests) usually show the best predictive performances. In this work, we propose a new sparse initialization technique for (potentially deep) multilayer perceptrons (MLP): we first train a tree-based procedure to detect feature interactions and use the resulting information to initialize the network, which is subsequently trained via standard gradient descent (GD) strategies. Numerical experiments on several tabular data sets show the benefits of this new, simple and easy-to-use method, both in terms of generalization capacity and computation time, compared to default MLP initialization and even to existing complex deep learning solutions. In fact, this wise MLP initialization raises the performances of the resulting NN methods to that of gradient boosting on tabular data. Besides, such initializations are able to preserve the sparsity of weights introduced in the first layers of the network throughout the training, which emphasizes that the first layers act as a sparse feature extractor (like convolutional layers in CNN).

## 1 Introduction

Neural networks are now widely used in many domains of machine learning, in particular when dealing with very structured data. They indeed provide state-of-the-art performances for applications with images or text. However, neural networks still perform poorly on tabular inputs, for which tree ensemble methods remain the gold standards (Grinsztajn et al., 2022). The goal of this paper is to improve the performances of the former by using the strengths of the latter.

**Tree ensemble methods**  Tree-based methods are widely used in the ML community, especially for processing tabular data. Two main approaches exist depending on whether the tree building process is parallel (e.g. Random Forest, RF, see Breiman, 2001b) or sequential (e.g. Gradient Boosting Decision Trees, GBDT, see Friedman, 2001). In these tree ensemble procedures, the final prediction relies on averaging predictions of randomized decision trees, coding for particular partitions of the input space. The two most successful and most widely used implementations of these methods are XGBoost and LightGBM (see Chen & Guestrin, 2016; Ke et al., 2017) which both rely on the sequential GBDT approach.

**Neural networks**  Neural Networks (NN) are efficient methods to unveil the patterns of spatial or temporal data, such as images (Krizhevsky et al., 2012) or texts (Liu et al., 2016). Their performance results notably from the fact that several architectures directly encode relevant structures in the input: convolutional neural networks (CNN, LeCun et al., 1995) use convolutions to detect spatially-invariant patterns in images, and recurrent neural networks (RNN, Rumelhart et al., 1985) use a hidden temporal state to leverage the natural order of a text. However, a dedicated *natural* architecture has yet to be introduced to deal with

---

*corresponding author: plutz@bu.edu

tabular data. Indeed, designing such an architecture would require to detect and leverage the structure of the relations between variables, which is much easier for images or text (spatial or temporal correlation) than for tabular data (unconstrained covariance structure).

**NN initialization and training** In the absence of a suitable architecture for handling tabular data, the Multi-Layer Perceptron (MLP) architecture (Rumelhart et al., 1986) remains the obvious choice due to its generalist nature. Apart from the large number of parameters, one difficulty of MLP training arises from the non-convexity of the loss function (see, e.g., Sun, 2020). In such situations, the initialization of the network parameters (weights and biases) are of the utmost importance, since it can influence both the optimization stability and the quality of the minimum found. Typically, such initializations are drawn according to independent uniform distributions with a variance decreasing w.r.t. the size of the layer (He et al., 2015). Therefore, one may wonder how to capitalize on methods that are inherently capable of recognizing patterns in tabular data (e.g., tree-based methods) to propose a new NN architecture suitable for tabular data and an initialization procedure that leads to faster convergence and better generalization performance.

## 1.1 Related works

How MLP can be used to handle tabular data remains unclear, especially since a corresponding prior in the MLP architecture adapted to the correlations of the input is not obvious, to say the least. Indeed, none of the existing NN architectures can consistently match the performance of state-of-the-art tree-based predictors on tabular data (Shwartz-Ziv & Armon, 2022; Gorishniy et al., 2021; and in particular Table 2 in Borisov et al., 2021).

**Self-attention architectures** Specific NN architectures have been proposed to deal with tabular data. For example, TabNet (Arik & Pfister, 2021) uses a sequential self-attention structure to detect relevant features and then applies several networks for prediction. SAINT (Somepalli et al., 2021), on the other hand, uses a two-dimensional attention structure (on both features and samples) organized in several layers to extract relevant information which is then fed to a classical MLP. These methods typically require a large amount of data, since the self-attention layers and the output network involve numerous MLP.

**Trees and neural networks** Several solutions have been proposed to leverage the correspondence between tree-based methods and NN, in order to develop more efficient models for processing tabular data. For example, TabNN (Ke et al., 2018) first trains a GBDT on the available data, then extracts a group of features per individual tree, compresses the resulting groups, and uses a tailored Recursive Encoder based on the structure of these groups (with an initialization based on the tree leaves). Therefore, TabNN employs pre-trained tree-based methods to design more efficient NN. Conversely, Sethi (1990) Brent (1991), and later Welbl (2014), Richmond et al. (2015) and Biau et al. (2019) propose to translate decision trees into very specific MLP (made of 3 layers) and use GD training to improve upon the original tree-based method. Such procedures can be seen as a way to relax and generalize the partition geometry produced by trees and their aggregation. To our knowledge, such translations have not been used to boost the training of *general* NN architectures.

## 1.2 Contributions

In this work, we propose a new method to initialize a potentially deep MLP for learning tasks with tabular data. Our method consists in first training a tree-based predictor (RF, GBDT or Deep Forest, see Section 2.1) and then using its translation into an MLP as initialization for the first two layers, the deeper ones being randomly initialized. With subsequent standard GD training, this procedure is shown to outperform the widely used uniform initialization of MLP (default initialization in Pytorch Paszke et al., 2019) as follows.

1. **Improved performances.** For tabular data, the predictive performances of the MLP after training are improved compared to MLP that use a random initialization. Our procedure also outperforms more complex deep learning procedures based on self-attention and is on par with classical tree-based methods (such as XGBoost).

2. **Faster optimization.** The optimization following a tree-based initialization is boosted in the sense that it enjoys a faster convergence towards a (better) empirical minimum: a tree-based initialization results in faster training of the MLP.

Initializing the first few layers of the MLP with the translation of the tree-based method and initializing randomly the deeper layers is the most successful initialization scheme that we experimented. This supports the idea that in our method, the (first) tree-based initialized layers act as relevant feature extractors that allow the MLP to detect correlations in the inputs. In this context, our approach is dedicated on improving the performance of standard MLP models; therefore it is conceptually different from pre-existing procedures also relying on the translation of tree-based models into NN: (Biau et al., 2019) aim at fine-tuning tree-based methods using a very specific neural network framework (made of only 3 layers). We, on the other hand, use tree-based methods to carefully initialize certain layers of a generic MLP, which is then substantially trained using standard GD strategies.

**Outline** In Section 2, we introduce the predictors in play and describe how tree-based methods can be translated into MLP. The core of our analysis is contained in Section 3, where we describe in detail the MLP initialization process and provide extensive numerical evaluations showing the benefits of this method.

## 2 EQUIVALENCE BETWEEN TREES AND MLP

Consider the classical setting of supervised learning in which we are given a set of input/output samples $\{(X_i, Y_i)\}_{i=1}^n$ drawn i.i.d. from some unknown joint distribution. Our goal is to construct a (MLP) function to predict the output from the input. To do so, we leverage the translation of tree-based methods into MLP.

### 2.1 PRESENTATION OF THE PREDICTORS IN PLAY

**Tree-based methods** We consider three different tree ensemble methods: Random Forests (RF), Gradient Boosting Decision Trees (GBDT) and Deep Forests (DF). They all share the same base component: the Decision Tree (DT, for details see Breiman et al., 1984). We call its terminal nodes *leaf nodes*, which correspond to the cells of the final tree partition. RF (Breiman, 2001a) is a predictor consisting of a collection of independently trained and randomized trees. Its final prediction is made by averaging the predictions of all its DT in regression or by a majority vote in classification. GBDT (Friedman, 2001) aims at minimizing a prediction loss function by successively aggregating DT that approximate the opposite gradient of that loss function (see Chen & Guestrin, 2016, for details on XGBoost). DF (Zhou & Feng, 2019) is a hybrid learning procedure in which random forests are used as elementary components (neurons) of a neural-network-like architecture (see Figure 5 and Appendix A for details).

**Multilayer Perceptron (MLP)** The multilayer perceptron is a predictor consisting of a composition of multiple affine functions, with (potentially different) nonlinear activation functions between them. Standard activation functions include, for instance, the rectified linear unit or the hyperbolic tangent. Deep MLP are a much richer class of predictors than tree-based methods which build simple partitions of the space and output piecewise constant predictions. Therefore, any of the tree-based models presented above can be approximated and in fact exactly rewritten as an MLP as follows.

### 2.2 AN EXACT TRANSLATION OF TREE-BASED METHODS INTO MLP

**From decision tree to 3-layer MLP** Recall that a decision tree codes for a partition of the input space in as many parts as there are leaf nodes in the tree. Given an input $x$, we can identify the leaf where $x$ falls by examining for each hyperplane of the partition whether $x$ falls on the right or left side of the hyperplane. The prediction is then made by averaging the outputs of all the training samples falling into the leaf of $x$. A DT can be thus translated into a highly sparse 3-layer MLP:

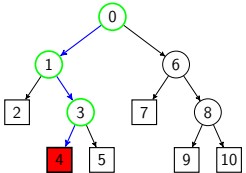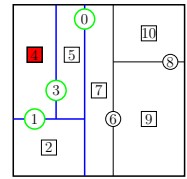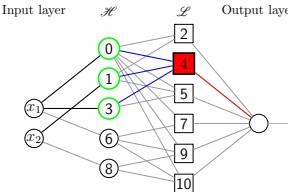

Figure 1: (from Biau et al., 2019) Illustration of a decision tree, its induced feature space partition and its corresponding MLP translation on a problem with 2 input variables.

1. The first layer contains a number of neurons equal to the number of hyperplanes in the partition, each neuron encoding by $\pm 1$ whether $x$ falls on the left or right side of the hyperplane.

2. The second layer contains a number of neurons equal to the number of leaves in the DT. Based on the first layer, it identifies in which leaf $x$ falls and outputs a vector with a single 1 at the leaf position and $-1$ everywhere else.

3. The last layer contains a single output neuron that returns the tree prediction. Its weights encode the average output of all training samples for each leaf of the tree.

This procedure is explained in detail and formally in Biau et al. (2019) and in Appendix B.

**From RF/GBDT to 3-layer MLP** Although RF and GBDT are constructed in different ways, they both average multiple DT predictions to give the final result. Thus, to translate a RF or a GBDT into an MLP, we simply turn each tree into a 3-layer MLP as described above, and concatenate all the obtained networks to form a wider 3-layer MLP. When concatenating, we set all weights between the MLP translations of the different trees to 0, since the trees do not interact with each other in predicting the target value for a new feature vector. The step in which the responses of the different trees are averaged can be combined with the third layer of the individual tree translations, resulting in a final MLP translation with a total of three layers.

**From Deep Forests to deeper MLP** A Deep Forest is a cascade of Random Forests. As such, it can be translated into an MLP containing the MLP translations of the different RF in cascade, resulting in a deeper and wider MLP (note that the obtained MLP has a number of layers that is a multiple of 3).Furthermore, in the Deep Forest architecture, the input vector is concatenated to the output of each intermediate layer. To mimic these skip connections in the MLP, we add additional neurons to each layer, except for the last three, which encode an identity mapping. Appendix A gives more insights into DF and their MLP translations. In particular, perfect translation of a DF suffers from numerical instabilities due to the replication of catastrophic cancellations (the deeper the DF, the greater their amplitude, cf Appendix D). This does not impact the sequel of the study, which relies on MLP approximations introduced in Section 2.2.

## 2.3 RELAXING TREE-BASED TRANSLATION TO ALLOW GRADIENT DESCENT TRAINING

As shown in the previous section, one can construct an MLP that exactly reproduces a tree-based predictor. However this translation involves (i) piecewise constant activation functions (sign) and (ii) different activation functions in a same layer (sign and identity when translating DF). These constraints can hinder the MLP training, which relies on GD strategies (requiring differentiability), as well as efficient implementation tricks, given that automatic differentiation libraries only support one activation function per layer. Therefore, given a pre-trained tree-based predictor (RF, GBDT or DF), we aim at relaxing its translation into a MLP, mimicking its behavior as closely as possible but in a compatible way with standard NN training.

**From tree-based methods to differentiable MLP** To do so, Welbl (2014); Biau et al. (2019) consider the differentiable $\tanh$ activation, well suited for approximating both the sign and

identity functions. Indeed, this can be achieved by multiplying or dividing the output of a neuron by a large constant before applying the function $\tanh$ and rescaling the result accordingly if necessary, i.e. for large enough $a, c > 0$, $\text{sign}(x) \approx \tanh(ax)$ and $x \approx c \tanh\left(\frac{x}{c}\right)$.

However, we cannot choose $a$ arbitrarily large as this would make gradients vanish during the network optimization (the function being flat on most of the space), and hinder training. We therefore introduce 4 hyper-parameters for the MLP encoding of any tree-based method that regulate the degree of approximation for the activation functions after the first, second and third layers of a decision tree translation, as well as for the identity mapping, respectively denoted by `strength01`, `strength12`, `strength23` and `strength_id`.

**Hyperparameter choice** The use of the tanh activation function involves extra hyperparameters. We study the influence of each one, by making them vary in some range (keeping the others fixed to $10^{10}$, resulting in an almost perfect approximation of the sign and identity functions), see Appendix D.1 for details. Our analysis shows that increasing the hyperparameters beyond some limit value is no longer beneficial (as the activation functions are already perfectly approximated) and, across multiple data sets, these limit values are similar. We also exhibit relevant search spaces that will allow us to find optimal HP values for each application.

## 3 A NEW INITIALIZATION METHOD FOR MLP TRAINING

In this section, we study the impact of tree-based initialization methods for MLP training when dealing with tabular data. The latter empirically proves to be always preferable to standard random initialization and makes MLP a competitive predictor for tabular data. Our code is publically available at `https://github.com/LutzPatrick/SparseTreeBasedInit`.

### 3.1 OUR PROPOSAL

Random initialization is the most common technique for initializing MLP prior to stochastic gradient training. It consists in setting all layer parameters to random values of small magnitude centered at 0. More precisely, all parameter values of the $j$-th layer are uniformly drawn in $[-1/\sqrt{d_j}, 1/\sqrt{d_j}]$ where $d_j$ is the layer input dimension; this is the default behaviour of most MLP implementations such as `nn.Linear` in PyTorch (Paszke et al., 2019).

We introduce new ways of initializing an MLP for learning with tabular data, by leveraging the recasting of tree-based methods in a neural network fashion:

- **RF/GBDT initialization.** First, a RF/GBDT is fitted to the training data and transformed into a 3-layer neural network, following the procedure described in Section 2. The first two layers of this network are used to initialize the first two layers of the network of interest. Thus, upon initialization, these first two layers encode the RF/GBDT partition. The parameters of the third and all subsequent layers are randomly initialized as described above. See Figure 7 in Appendix C for an illustration.

- **DF initialization.** Similarly as above, a Deep Forest (DF) using $\ell$ forest layers is first fitted to the training data. The first $3\ell-1$ layers of the MLP are then initialized using the first $3\ell - 1$ layers of the MLP encoding of this pre-trained DF. The parameters of the $3\ell$-th and all subsequent layers are randomly initialized as explained above.

These tree-based initialization techniques may seem far-fetched at first glance, but they are actually consistent with recent approaches to adapting Deep Learning models for tabular data. The key to interpreting them is to think of the first (tree-based initialized) layers of the MLP as a feature extractor that produces an abstract representation of the input data (in fact, this is a vector encoding the tree-based predictor's space partition in which the observation lies). The subsequent randomly initialized layers, once trained, then perform the prediction task based on this abstract representation.

## 3.2 Experimental setup

**Datasets & learning tasks**   We compare prediction performances on a total of 10 datasets: 3 regression datasets (Airbnb, Diamonds and Housing), 5 binary classification datasets (Adult, Bank, Blastchar, Heloc, Higgs) and 2 multi-class classification datasets (Covertype and Volkert). We mostly chose data sets that are used for benchmarking in relevent literature: Adult, Heloc, Housing, Higgs and Covertype are used by Borisov et al. (2021) and Bank, Blastchar and Volkert are used by Somepalli et al. (2021). Moreover, we add Airbnb and Diamonds to balance the different types of prediction tasks. The considered datasets are all medium-sized (10–60k observations) except for Covertype and Higgs (approx. 500k observations). Details about the datasets are given in Appendix E.1.

**Predictors**   We consider the following tree-based predictors: Random Forest (RF), Deep Forest (DF, Zhou & Feng, 2017) and XGBoost (denoted by GBDT, Chen & Guestrin, 2016). The latter usually achieves state-of-the-art performances on tabular data sets (see, e.g., Shwartz-Ziv & Armon, 2022; Gorishniy et al., 2021; Borisov et al., 2021). We also consider deep learning approaches: MLP with default uniform initialization (MLP rand. init.) or tree-based initialization (resp. MLP RF init., MLP GBDT init. and MLP DF init.); and a transformer architecture SAINT Somepalli et al. (2021). This complex architecture is specifically designed for applications on tabular data and includes self-attention and inter-sample attention layers that extract feature correlations that are then passed on to an MLP. For regression and classification tasks, we use the mean-squared error (MSE) and cross-entropy loss for NN training, respectively. We choose SAINT as a baseline model as it is reported to outperform all other NN predictors on most of our data sets (all except Airbnb and Diamonds, see Borisov et al., 2021; Somepalli et al., 2021).

**Parameter optimization**   All NN are trained using the Adam optimizer (Kingma & Ba, 2014). All hyper-parameters (HP) are determined empirically using the optuna library (Akiba et al., 2019) for Bayesian optimization.

For most HP, we use the default search spaces of Borisov et al. (2021). For all HP tuning the tree-to-MLP translation, we have identified relevant search spaces (see Appendix D.1). An overview of all search spaces used for each method and the HP selected for experimental protocol P2 can be found in Appendix E.5. The quantity minimized during HP tuning is the model's validation loss, and the smallest validation loss that occurred during training for MLP-based models.

## 3.3 A better MLP initialization for a better optimization

In this subsection, the optimization of standard MLP is shown to benefit from the proposed initialization technique. Experiments have been conducted on 6 out of the 10 data sets.

**Experimental protocol 1 (P1)**   To obtain comparable optimization processes, we ensure that all MLP-related hyper-parameters (width, depth, learning rate), are identical for all the

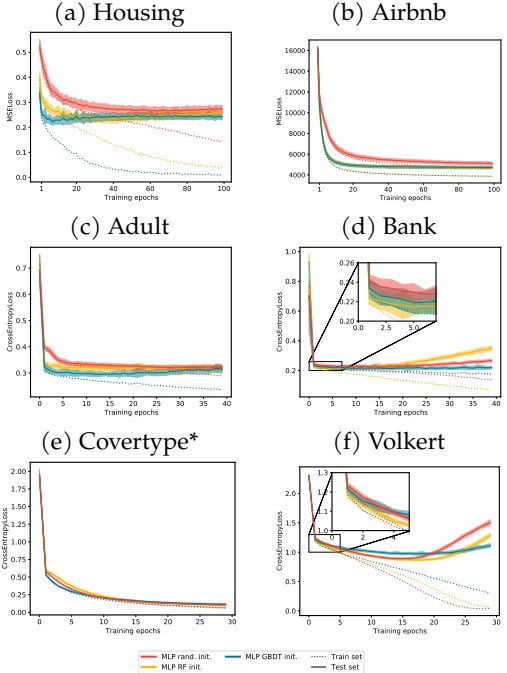

Figure 2: Optimization behaviour of randomly, RF and GBDT initialized MLP evaluated over a 5 times repeated (stratified) 5-fold of each data set, according to Protocol P1. The lines and shaded areas report the mean and standard deviation. *evaluation on a single 5-fold cross validation.

MLP regardless of the initialization scheme. These HP are chosen to maximize the predictive performance of the standard randomly initialized MLP. All HP related to the ini-

tialization technique (HP for the tree-based predictors and their translation) are optimized independently for each tree-based initializations.

**Results** Figure 2 shows that for most data sets, the use of tree-based initialization methods for MLP training provides a *faster convergence* towards *a better minimum* (in terms of generalization) than random initialization. This is all the more remarkable since Protocol P1 has been calibrated in favor of random initialization. Among tree-based initializers, GBDT initializations outperform or are on par with RF initializations in terms of the optimization behavior on all regression and binary classification problems. However, for multiclass classification problems, the advantages of tree-based initialization seem to be limited. This is probably due to the fact that the MLP architecture at play is tailored for random initialization, being thus too restrictive for tree-based initializers. Experiments presented in Appendix E.3 with fixed arbitrary widths corroborate this idea: in this case, the RF initialization is beneficial for the optimization process. For the Adult, Bank, and Volkert data sets, Figure 2 also shows the performance of each method at initialization. None of these procedures leads to a better MLP performance at initialization (due to both the non-exact translation from trees to MLP and to the additional randomly initialized layers), but rather help guiding the MLP in its learning process.

### 3.4 A better MLP initialization for a better generalization

In this subsection, tree-based initialization methods are shown to systematically improve the predictive power of neural networks compared to random initialization. We compare our procedure to the predictors described in Section 3.2, but also to 3 other NN techniques: one close to the default uniform initialization (Xavier init., see Glorot & Bengio, 2010), one using random orthonal matrices (LUSV init., see Mishkin & Matas, 2015) and the winning ticket lottery strategy (WT prun., see Frankle & Carbin, 2018), which is a pruning method used during training to end up with a sparse NN. The reader may refer to Appendix E.4.1 for more details about these three techniques.

**Experimental protocol 2 (P2)** Each MLP is trained on 100 epochs, but with HP tuned depending on the initialization technique. For maximum comparability, the optimization budget is strictly the same for all methods (100 "optuna" iterations each, where one optuna iteration consists of one hold-out validation). In particular, when using a tree-based initializer, we use 25 HP optimization iterations to find optimal HP for the tree-based predictor, fix these HPs, and then use the remaining 75 iterations to determine optimal HP for the MLP. For all NN approaches, the model with the best performance on the validation set during training is kept (using the classical *early-stopping* procedure). Performances are measured via the MSE for regression, the AUROC score (AUC) for binary classification and the accuracy (Acc.) for multi-class classification, averaging 5 runs of 5-fold cross-validation.

**Results** Table 1 shows that RF or GBDT initialization strictly outperform random initialization, in terms of final generalization performance, for all data sets except Covertype (for which performances are similar). They also systematically achieve better results than the LUSV and Xavier init. and are better on all but 2 datasets than the WT pruning procedure which is a more refined procedure. Additionally, the MLP using both RF and GBDT initialization techniques outperform SAINT on all medium-sized data sets and fall short on large data sets (Higgs and Covertype).

Despite its simplicity , the proposed method (based on RF or GBDT) is on par with GBDT on half of the data sets, ranking MLP as relevant predictors for tabular data. Note that the GBDT used for initialization of the MLP is way less powerful than the best one found here (see details in Tables 10 and 11). This shows that our procedure produces, with a relatively low initialization cost, powerful MLP relevant for tabular data. Among the tree-based initializers, RF is on par with or outperforms GBDT initialization on all data sets but Housing. DF initialization, for its part, cannot compete with RF and GBDT initialization, despite showing some improvement over the random one (except for Covertype and Volkert). This underlines that injecting prior information via tree-based methods into the first layers of a MLP is, among all the aforementioned methods, the best way to improve its performance.

| Data set / Model | Housing | Airbnb | Diamonds | Adult | Bank | Blastchar | Heloc | Higgs | Covertype | Volkert |
|---|---|---|---|---|---|---|---|---|---|---|
| | MSE ↓ | MSE ↓ (x10³) | MSE ↓ (x10⁻³) | AUC ↑ (in %) | AUC ↑ (in %) | AUC ↑ (in %) | AUC ↑ (in %) | AUC ↑ (in %) | Acc. ↑ (in %) | Acc. ↑ (in %) |
| Random Forest | 0.263±0.009 | 5.39±0.13 | 9.80±0.35 | 91.6±0.3 | 92.8±0.3 | **84.5±1.2** | 91.3±0.6 | 80.4±0.1 | 83.6±0.1 | 64.2±0.3 |
| GBDT | **0.208±0.010** | **4.71±0.15** | **7.38±0.28** | **92.7±0.3** | **93.3±0.3** | 84.7±1.0 | **92.1±0.4** | 82.8±0.1 | **97.0±0.0** | 71.3±0.4 |
| Deep Forest | 0.225±0.008 | **4.68±0.16** | 8.23±0.29 | 91.8±0.3 | 92.9±0.2 | 83.7±1.2 | 90.3±0.5 | 81.2±0.0* | 92.4±0.1* | 66.3±0.4 |
| MLP rand. init. | 0.258±0.011 | 5.07±0.16 | 15.5±12.5 | 90.5±0.4 | 91.0±0.5 | 81.4±1.2 | 80.1±0.1 | 83.2±0.3 | 96.7±0.0 | 72.2±0.4 |
| MLP Xavier init. | 0.263±0.012 | 5.05±0.17 | 12.4±6.19 | 90.5±0.5 | 90.8±0.5 | 81.7±1.3 | 79.9±1.1 | 82.8±0.1 | 96.8±0.0 | 72.1±0.4 |
| MLP LUSV init. | 0.295±0.018 | 4.99±0.14 | 14.1±5.00 | 90.5±0.5 | 90.2±0.5 | 84.3±1.2 | 79.9±0.9 | 81.7±0.1 | 96.5±0.0 | 70.8±0.5 |
| MLP WT prun. | 0.248±0.011 | 5.26±2.11 | 9.83±5.07 | 90.6±0.4 | 90.9±0.5 | 84.4±1.2 | 89.6±0.7 | 82.9±0.1 | 97.0±0.0 | 71.5±0.4 |
| MLP RF init. | 0.222±0.009 | 4.66±0.16 | 7.93±0.22 | 92.1±0.3 | 92.4±0.4 | 84.4±1.2 | **91.7±0.4** | 83.6±0.1 | 96.7±0.0 | **74.1±0.4** |
| MLP GBDT init. | **0.206±0.007** | 4.70±0.09 | 8.15±0.35 | 92.2±0.3 | 92.5±0.3 | 84.6±1.2 | 91.5±0.6 | 83.0±0.0 | 96.2±0.0 | 73.5±0.5 |
| MLP DF init. | 0.234±0.016 | **4.81±0.13** | 8.28±0.24 | 91.9±0.4 | 92.2±0.3 | 84.2±1.0 | 91.4±0.6 | 83.3±0.1* | 94.5±0.3* | 71.3±0.5 |
| SAINT | 0.258±0.011 | **4.81±0.15** | 17.7±3.83 | 91.6±0.3 | 92.2±0.4 | 84.0±0.8 | 90.2±0.7 | **83.7±0.1*** | 96.6±0.1* | 70.1±0.4 |

Table 1: Best scores and their standard deviations for Protocol 2. For each data set, predictors performing at least as well as the best over all (resp. best DL) score up to its standard deviation are highlighted in **bold** (resp. underlined). All scores are based on a 5 times repeated (stratified) 5-fold cross validation. For each model, HP have been chosen via the "optuna" library with 100 iterations. See Appendix E.4.2 for a comparison with literature results. *score based on a simple 5-fold cross val.

The interested reader may find a comparison of the optimization procedures of all MLP methods and SAINT (Figure 13) and tables summarizing all HP (Tables 10 and 11) in Appendix E.5.2. We remark that tree-based initializers generally bring into play wider networks with similar depths (fixed width of 2048 and adaptive depth between 4 and 10) compared to MLP with default initialization. Yet, for most data sets, the overall procedure is computationally more efficient than state-of-the-art deep learning architectures like SAINT, both in terms of number of parameters and training time (see Tables 6-8 in Appendix E.4).

## 3.5 ANALYZING KEY ELEMENTS OF THE NEW INITIALIZATION METHODS

**Influence of the MLP width** We mainly use standard search spaces from (Borisov et al., 2021) to determine the optimal hyper-parameters for each model. However, the MLP width is an exception to this. The standard search spaces used in the literature usually involve MLP with a few hundred neurons per layer (e.g. up to 100 neurons in Borisov et al., 2021); yet, in this work, we consider MLP with a width up to 2048 neurons. Large MLP are actually very beneficial for tree-based initialization methods as they allow the use of more expressive tree-based models in the initialization step.

Figure 3 compares the performance of an MLP with random/GDBT initializations and various widths. There is no gain in prediction by using wider (thus more complex) NN, when randomly initialized. This is corroborated by the results of Table 4: for all regression and binary classification data sets, the performance of our (potentially much wider) MLP with random initialisation is consistently close to the literature values, and only increases for multi-class classification tasks. However, an MLP initialized with GBDT significantly benefits from enlarging the NN width (justifying a fixed width of 2048 for tree-based initialized MLP). This confirms the idea that tree-based initialization helps revealing relevant features to the MLP, all the more as the width increases, and by doing so, boosts the MLP performance after training.

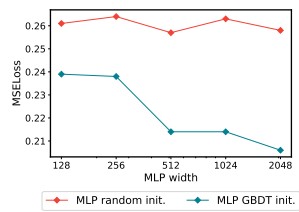

Figure 3: Influence of width on the generalization performance for random and GBDT initializations. Mean values over 5 times repeated 5-fold cross-validation on Housing.

**Performance of the initializer** Another interesting step in unraveling the essence of the new initialization method is to understand which characteristics of a tree-based model are relevant to its success as an initializer. Undoubtedly, its predictive accuracy plays an important role, but does this aspect alone suffice to characterize the success of the new initialization method? Figure 14 compares the predictive performance of different RF/GBDT initializers and the performance of the respective MLP after training. As the figure illustrates, a better performance of the tree-based predictor used for initialization does not always lead

to a better performance of the MLP after training (see Airbnb and Volkert). This observation suggests that other aspects, such as the expressiveness of the feature interactions captured by the initializer, the structure it induces on the MLP or the weight distributions of the initializer, must also play a significant role in the initialization method's success.

**MLP sparsity**   Finally, we investigate the structure that tree-based initialization induces on the MLP *after* training. Figure 4 shows the weight distributions of the three first and the last layers before and after MLP training, for random, RF and GBDT initializations on Housing (see Appendix E.7 for more data sets). It indicates that the weight distribution on the first two layers change significantly during training when the MLP is randomly initialized: the weights are uniformly distributed at epoch 0 but appear to be Gaussian after training. When RF or GBDT initializers are used instead, the weights of the first two layers are sparsely distributed at epoch 0 by construction, and their distribution is preserved during training (notice the logarithmic y-axis for these plots in Figure 4). Note that the (uniform) distribution of the weights in other layers is also preserved through training (third and last lines of Figure 4). This means that our initialization technique, in combination with SGD optimization strategies, introduces an *implicit regularization* of the NN optimization: the sparse structure of the initialisation (on first layers) is maintained. This is very similar to the CNN architecture (constrained by design), a very successful class of NN designed for image processing. Besides, the weight distributions are not squeezed towards zero during learning when sparse initialization is used, preventing poor generalization performances according to previous works (Neal, 2012; Blundell et al., 2015).

Looking at Figure 4, one may get the impression that the weights in the first layers remain unchanged during GD training, and that ultimately no learning takes place in these layers. However, numerical experiments (see Appendix E.4.3) show that the weights of *all* layers are modified during learning; the first two layers actually undergo the greatest changes. As such, training the RF/GBDT-initialized layers via GD strategies is essential for the success of the new initialization method.

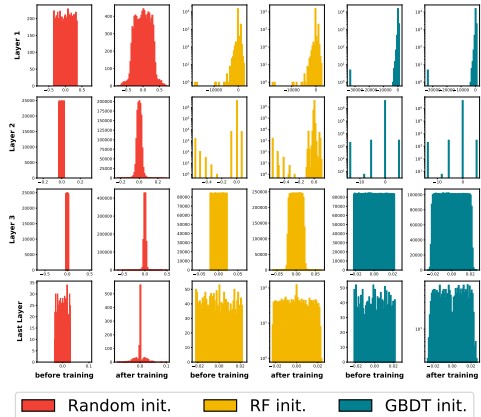

Figure 4: Histograms of the first three and last layers' weights before and after the MLP training on Housing. Comparison between random, RF and GBDT initializations.

## 4   Conclusion and Future work

This work builds upon the permeability that exists between tree methods and neural networks, in particular how the former can help the latter during training, with tabular inputs. We proposed new methods for smartly initializing the first layers of standard MLP using pre-trained tree-based methods. The sparsity of this initialization is preserved during training, which shows that it encodes relevant correlations between the data features. Among deep learning methods, such initializations of MLP always improve the performance compared to the widely used random initialization, and provide an easy-to-use and more efficient alternative to SAINT (attention-based method) for tabular data. The performance of this wisely-initialized MLP is remarkably approaching that of XGBoost, which so far reigns supreme for learning tasks on tabular data.

**Limitations & future work**   While our procedure is quite generic, some restrictions are noticeable. First, our analysis only allows to initialize neural networks with $\tanh$ activation functions; removing this limitation by considering ReLU is a good avenue for future work. Besides, while quite reasonable, our initialization is more time-consuming than the random (default) one. Moreover, we need to further investigate the benefits of our initialization method on very large data sets. Finally, another interesting direction could be using the efficient hyperparameter search in tree-based methods to automatically determine a good default NN architecture.

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

## A    Details on Deep Forest (DF) and its translation

The layers of DF are composed of an assortment of Breiman's Random Forests and Completely-Random Forests (CRF, Fan et al. (2003)) and are trained one after another in a cascade manner. At a given layer, the outputs of all forests are concatenated, together with the raw input data. This new vector serves as input for the next DF layer. This process is repeated for each layer and the final output is obtained by averaging the forest outputs of the best layer (without raw data).

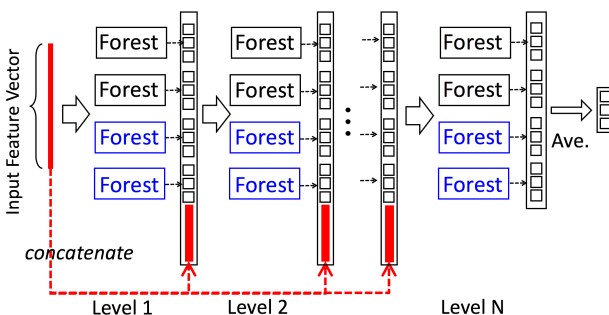

Figure 5: Illustration of the Deep Forest cascade structure for a classification problem with 3 classes. Each level of the cascade consists of two Breiman RFs (black) and two completely random forests (blue). The original input feature vector is concatenated to the output of each intermediate layer. Figure taken from (Zhou & Feng, 2017).

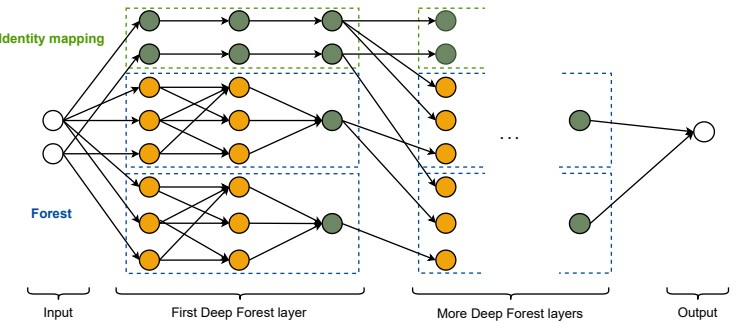

Figure 6: Illustration of the MLP translation of a Deep Forest. Yellow nodes use the $x \mapsto 2.\mathbb{1}_{\{x>0\}} - 1$ activation function and green nodes use the identity activation function.

## B    Details of the translation of a decision tree into an MLP

Recall that a decision tree codes for a partition of the input space in as many parts as there are leaf nodes in the tree. To know in which partition cell an input feature vector $x \in \mathbb{R}^d$ falls into, we move in the tree from the root to the corresponding leaf using simple rules: at each $m$-th inner node, $x$ is passed onto the left child node if its $i_m$-th coordinate is less than or equal to some threshold $t_m$, and to the right child node otherwise. The decision rule at each inner node of the tree introduces a split of the feature space into two subsets $\mathcal{H}_m^- = \{x \in \mathbb{R}^d \mid x^{(i_m)} \le t_m\}$ and $\mathcal{H}_m^+ = \{x \in \mathbb{R}^d \mid x^{(i_m)} > t_m\}$. Consistent with how the MLP translation works, we intentionally define $\mathcal{H}_m^-$ and $\mathcal{H}_m^+$ such that at each inner node $m$, $\mathcal{H}_m^- \cup \mathcal{H}_m^+ = \mathbb{R}^d$. Let $N$ be the number of inner nodes of the decision tree; note that the decision tree has exactly $N + 1$ leaf nodes, since it is by definition a complete binary tree, see Figure 1 for an illustration. For a leaf node $\ell \in \{1, \dots, N + 1\}$ of the tree, let $\mathcal{P}_\ell^- \subset \{1, \dots, N\}$ (respectively $\mathcal{P}_\ell^+$) be the set of all inner nodes whose left (respectively

right) subtree contains $\ell$, that is, $\mathcal{P}_\ell^+ \cup \mathcal{P}_\ell^-$ is the set of all parent nodes of $\ell$. Then, the decision tree sorts an observation $x \in \mathbb{R}^d$ into its leaf $\mathcal{R}_\ell$ if and only if

$$x \in \mathcal{R}_\ell = \left( \bigcap_{m \in \mathcal{P}_\ell^-} \mathcal{H}_m^- \right) \cap \left( \bigcap_{m \in \mathcal{P}_\ell^+} \mathcal{H}_m^+ \right). \tag{1}$$

In fact, $\{\mathcal{R}_\ell\}_{\ell \in \mathcal{L}}$ is the feature space partition coded by the tree, see Figure 1 for an example. Finally, the tree returns the average response of all training samples that fall into the same leaf as the input data; let us call $a_\ell$ the average response of all training samples in $\mathcal{R}_\ell$. The final prediction of the decision tree $g$ can therefore be expressed as

$$g(x) = \sum_{\ell=1}^{N+1} a_\ell \mathbb{1}_{\{x \in \mathcal{R}_\ell\}}.$$

Let us now explore how an MLP can be designed to reproduce the prediction of a decision tree. Consider an MLP of depth 3 with $N$ neurons on the first layer. For each inner node $m \in \{1, \ldots, N\}$, the $m$-th neuron of the first layer indicates on which side of the split introduced by this inner node a given feature vector lies: it equals $-1$ if the feature vector lies in $\mathcal{H}_m^-$ and $+1$ if it lies in $\mathcal{H}_m^+$. This can be achieved applying the following affine transformation and a sign activation function to the feature vector,

$$A_1 : x \in \mathbb{R}^d \mapsto x^{(i_m)} - t_m \quad \text{and} \quad \varphi_1 : x \mapsto \begin{cases} -1 & \text{if } x \leq 0 \\ 1 & \text{if } x > 0. \end{cases}$$

The second layer of the 3-layer MLP has $N+1$ neurons. For each leaf node $\ell \in \{1, \ldots, N+1\}$, the $\ell$-th neuron of the second layer indicates whether a given feature vector $x \in \mathbb{R}^d$ lies in $\mathcal{R}_\ell$ or not: it equals $+1$ if $x \in \mathcal{R}_\ell$ and $-1$ if $x \notin \mathcal{R}_\ell$. Using equation (1), this can be achieved by applying the following affine transformation and a sign activation function to the output of the first layer,

$$A_2 : x \in \mathbb{R}^N \mapsto \sum_{m \in \mathcal{P}_\ell^+} x^{(m)} - \sum_{m \in \mathcal{P}_\ell^-} x^{(m)} - \left| \mathcal{P}_\ell^+ \cup \mathcal{P}_\ell^- \right| + \frac{1}{2} \quad \text{and} \quad \varphi_2 : x \mapsto \begin{cases} -1 & \text{if } x \leq 0 \\ 1 & \text{if } x > 0. \end{cases}$$

The last layer of the MLP contains a single output neuron that returns the tree prediction. Using the output of the second layer, this can be achieved by applying the following affine transformation and an identity activation function,

$$A_3 : x \in \mathbb{R}^{N+1} \mapsto \frac{1}{2} \left( \sum_{\ell=1}^{N+1} x^{(\ell)} a_\ell + \sum_{\ell=1}^{N+1} a_\ell \right) \quad \text{and} \quad \varphi_3 : x \mapsto x \tag{2}$$

where $a_\ell$ is the average response of all training samples in $\mathcal{R}_\ell$. Note that $\{a_\ell\}_{\ell=1}^{N+1}$ is a set of real numbers in regression problems and a set of probability vectors representing class distributions in classification problems. An illustration of the MLP translation of a decision tree is shown in Figure 1. This translation procedure is explained, for example, in Biau et al. (2019) with more details.

## C    ILLUSTRATION OF OUR INITIALISATION METHOD

We provide below an illustration (Figure 7) showing how the whole MLP is initialised using both the tree-based method for the first layers and the random initialisation for the deeper layers.

## D    DETAIL ON THE MLP TRANSLATION ACCURACY

Recasting a Deep Forest into a deep MLP using our method may suffer from numerical instabilities altering the predictive behaviour. This is due to a phenomenon of catastrophic cancellation, more likely to occur with deep MLP translations. This is explained in the following section.

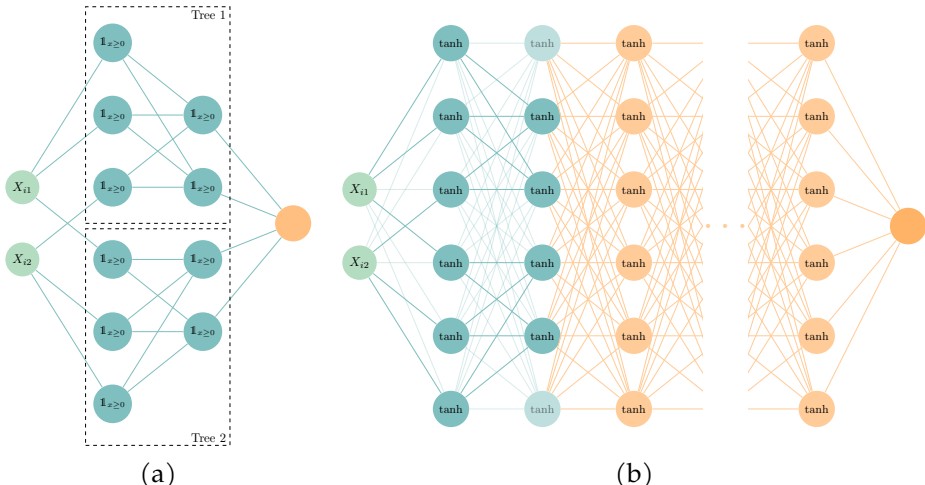

Figure 7: Illustration of the initialization technique on an MLP with 2 inputs and 1 output. In (a), a pre-trained tree-based method composed of 2 trees is represented in a NN fashion involving indicator functions as activation functions. In (b), an MLP of arbitrary depth and involving tanh activation functions is represented at initialization: the weights of the first two layers are initialized using the information captured in (a) (note that all connections marked in transparent blue are initialized to 0). The weights of the subsequent layers are randomly initialized (orange).

### D.1 On the choice of hyper-parameters

In Section 2.3, four hyper-parameters were introduced to approximate the sign and identity functions through the layers of an elementary MLP. We address here the choice of the HPs and propose an optimal range for these parameters in the sense that they are as small as possible while guaranteeing a faithful MLP translation.

We focus on the analysis of deep forest translation, as the structure of all other tree-based methods can be seen as a truncated variant of a deep forest. The deep forest is trained and translated into an MLP on each data set (see Section 2) for different values of the HPs. To identify the influence of each HP, we make them vary in some range while the other three HPs are fixed to $10^{10}$, resulting in an almost perfect approximation of the respective sign and identity functions. Figure 8 shows the predictive performance of a deep forest and its MLP translation playing with different HPs.

Figure 8 shows in particular that

(i) increasing the HPs beyond some limit value is no longer beneficial as the activation functions are already perfectly approximated;

(ii) across multiple data sets, these limit values are similar.

One could note that the coefficients in the first layer of a decision tree translation should be of a larger order of magnitude than those corresponding to the other activation functions to achieve an accurate translation. To give some insight into why this is the case, recall that the $m$-th neuron of the first layer determines whether the input vector belongs to $\mathcal{H}_m^-$ or $\mathcal{H}_m^+$, and note that its outputs can be of arbitrarily small size because the vector can be arbitrarily close to the decision boundaries. Note also that an MLP translation would better compromise on translation accuracy to ensure sufficient gradient flow. Based on these observations, we remark that choosing the HP of the following orders allows for maximum gradient flow while still providing an accurate translation: `strength01` $\in [1, 10^4]$, `strength12` $\in [10^{-2}, 10^2]$, `strength23` $\in [10^{-2}, 10^2]$ and `strength_id` $\in [10^{-2}, 10^2]$. This will actually help us later on to calibrate the search spaces when empirically tuning these HPs for each data set.

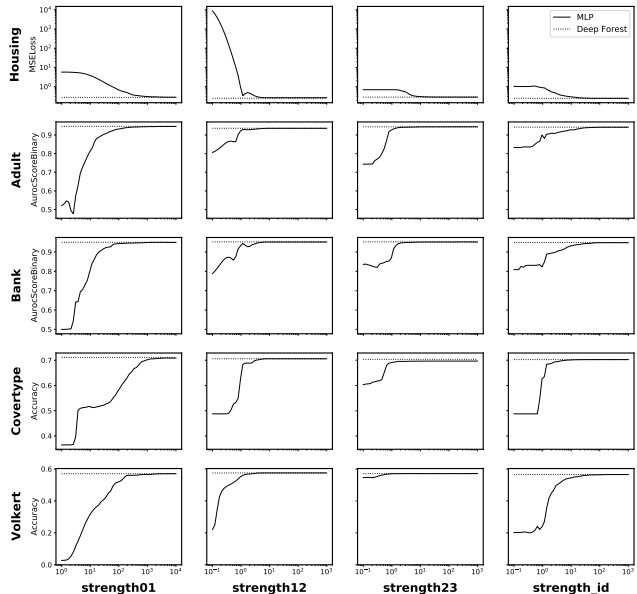

Figure 8: Comparison of the performance of a trained deep forest and its neural network encoding. Deep forest architecture: maximal depth of 8 per tree, 8 trees per forest, 1 forests per layer, 3 layers.

### D.2 A FUNDAMENTAL NUMERICAL INSTABILITY OF THE NEURAL NETWORK ENCODING

The encoding of a decision tree by a neural network proposed in Section 2.3 is numerically unstable, i.e., it does not necessarily yield the same result as the tree itself, even when using the original, non-approximated activation functions. This is the result of a catastrophic cancellation that occurs within the MLP translation. The term catastrophic cancellation describes the remarkable loss of precision that occurs when two nearly equal numbers are numerically subtracted. For example, take the numbers $a = 1$ and $b = 10^{-10}$, and perform the computation $(a + b) - a$ on a machine with limited precision, say to 8 significant digits. The machine will return $(a + b) - a = 1 - 1 = 0$, although this result is clearly not correct. This phenomenon occurs in the third layer of the MLP encoding, see equation (2). The two sums calculated in this layer are almost equal in magnitude but have opposite signs, resulting in a catastrophic cancellation that has a greater impact the more partitions of the input space the decision tree uses, i.e. the deeper it is.

Figure 9 illustrates the effect of this phenomenon, comparing the mean approximation error between a simple decision tree and its neural network encoding on the airbnb data set. In Figure 9a, the result at the output layer of the tree was replaced by the exact training mean of the corresponding decision tree partition, compensating for the catastrophic cancellation. No such compensation was done for Figure 9b. This shows the grave implications of this instability: the mean error grows exponentially with the depth of an individual tree.

Although the errors introduced by this phenomenon may not be large for a given decision tree, they might accumulate when several such trees are composed, for example in Random or Deep Forests. Figure 10 compares the mean approximation error between Random/Deep Forests of different complexities and their corresponding neural net encoding on the Airbnb data set. It shows that the composition of several trees in a cascade manner, as performed by the Deep Forest, leads to a stronger amplification of their individual inaccuracies than the parallel composition of trees, as performed by the Random Forest. This result is to be expected because decision trees composed in parallel do not influence each other's predictions, whereas in a cascade architecture the results of the first layer of decision trees affect the input of the subsequent layers and inaccuracies can thus develop stronger effects.

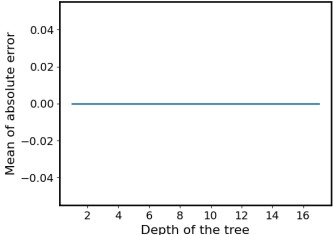 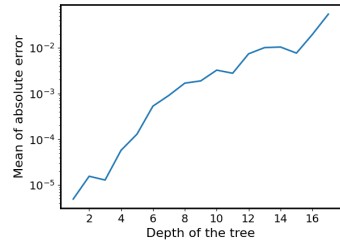

(a) replacing the output layer's result with the exact training mean of the corresp. tree partition

(b) using the output layer's result with catastrophic cancellation

Figure 9: Illustration of the fundamental numerical instability of the decision tree encoding.

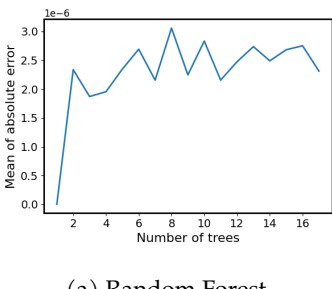 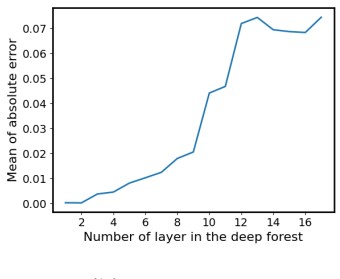

(a) Random Forest.

(b) Deep Forest

Figure 10: Effects of numerical instabilities on more complex tree-based predictors. Airbnb data set. Random Forests are composed of trees of depth 7. Deep forest architecture: tree depth of 7, 5 trees per forest, 1 forest per layer and a variable number of layers.

We note that this catastrophic cancellation can be easily circumvented by introducing an additional layer. If this maps the output of the second layer from $\{-1, 1\}$ to $\{0, 1\}$, the last layer could then simply multiply each of these outputs by the average response of a partition set. However, Figure 10 also shows that the error introduced by the catastrophic cancellation remains relatively small, except for deep forests with many layers. Therefore, we did not immediately address this issue and planned to fall back on this analysis if the MLP coding did not produce the expected results later in our analysis. However, this somewhat imprecise MLP coding worked well for all our purposes.

## E  SUPPLEMENTS TO NUMERICAL EVALUATIONS

### E.1  DATA SETS

**Data sets description**    In the sequel, we run numerical experiments on 10 real-world, heterogeneous, tabular data sets, all but two of which have already been used to benchmark deep learning methods, see Borisov et al. (2021); Somepalli et al. (2021). The chosen data sets represent a variety of different learning tasks and sample sizes. Tables 2 & 3 respectively give links to the platforms storing the data sets (four of them are available on the UCI Machine Learning Repository, Dua & Graff, 2017) and an overview of their main properties.

The Housing data set contains U.S. Census household attributes and the associated learning task is to predict the median house value for California districts (Pace & Barry, 1997). The Airbnb data set is provided by the company itself and holds attributes on different Airbnb listings in Berlin, such as the location of the apartment, the number of reviews, etc. The goal is to predict the price of each listing. Similarly, the diamond data set contains characteristics of different diamonds (e.g., carat weight or cut quality), and the goal is to predict the price of a diamond. The Adult data set contains Census information on adults (over 16-year olds) and its prediction task is to determine whether a person earns over $50k a year. The Bank

| Data set | Link |
|---|---|
| Housing | Scikit-learn |
| Airbnb | Inside Airbnb |
| Diamond | OpenML |
| Adult | UCI Machine Learning Repository |
| Bank | UCI Machine Learning Repository |
| Blastchar | Kaggle |
| Heloc | FICO |
| Higgs | UCI Machine Learning Repository |
| Covertype | UCI Machine Learning Repository |
| Volkert | AutoML |

Table 2: Links to data sets.

| | Housing | Airbnb | Diamonds | Adult | Bank | Blastchar | Heloc | Higgs | Covertype | Volkert |
|---|---|---|---|---|---|---|---|---|---|---|
| Dataset size | 20 640 | 119 268 | 53 940 | 32 561 | 45 211 | 7 043 | 9 871 | 550 000 | 581 012 | 58 310 |
| # Num. features | 8 | 10 | 6 | 6 | 7 | 3 | 21 | 27 | 44 | 147 |
| # Cat. features | 0 | 3 | 3 | 8 | 9 | 17 | 2 | 1 | 10 | 0 |
| Task | Regr. | Regr. | Regr. | Classif. | Classif. | Classif. | Classif. | Classif. | Classif. | Classif. |
| # Classes | - | - | - | 2 | 2 | 2 | 2 | 2 | 7 | 10 |

Table 3: Main properties of the data sets.

data set is related with direct marketing campaigns (phone calls) of a Portuguese banking institution, the classification goal is to predict whether the client will subscribe a term deposit. The Blastchar data set features information on customers of a fictional company that provides phone and internet services. The classification goal is to predict whether a customer cancels their contract in the upcoming month. The Heloc data set contains personal and credit record information on people that recently took on a line of credit, the classification task being to predict whether they will repay this credit within 2 years. On the Higgs data set (Baldi et al., 2014), the classification problem is to distinguish between signal processes that produce Higgs bosons and background processes that do not. For this purpose, it contains kinematic properties measured by the particle detectors in the accelerator that have been produced using Monte Carlo simulations. The Covertype data set contains cartographic variables on forest cells and it's task is to predict the forest cover type. Finally, for the Volkert data set, different patches of the same size have been cut from images that belong to 10 different landscape scenes (coast, forest, mountain, plain, etc.). Each observation contains visual descriptors of one patch, the goal of this classification problem is to find the landscape type of the original picture.

### E.2 IMPLEMENTATION DETAILS

RFs are implemented using `sklearn`'s `RandomForestRegressor` and `RandomForestClassifier` classes with default configuration for all parameters that are not mentioned explicitly. DFs are implemented using the `ForestLayer` library (Zhou & Feng, 2017) and GBDTs are implemented using the `XGBoost` library (Chen & Guestrin, 2016). MLPs are implemented and trained with `pytorch`, using the mean-squared error and the cross entropy as objective function for regression and classification problems respectively. The SAINT model is implemented using the library provided by Somepalli et al. (2021).

All methods are trained on a 32 GB RAM machine using 12 Intel Core i7-8700K CPUs, and one NVIDIA GeForce RTX 2080 GPU when possible (only the GDBT and MLP implementations including SAINT use the GPU). Hyper-parameter searches are parallelized on up to 4 of these machines.

**Hyper-parameter optimization** We tune all hyper-parameters using the `optuna` library (Akiba et al., 2019) with a fixed number of iterations for all models. In this context, an *iteration* corresponds to a set of hyper-parameters whose performance is evaluated with

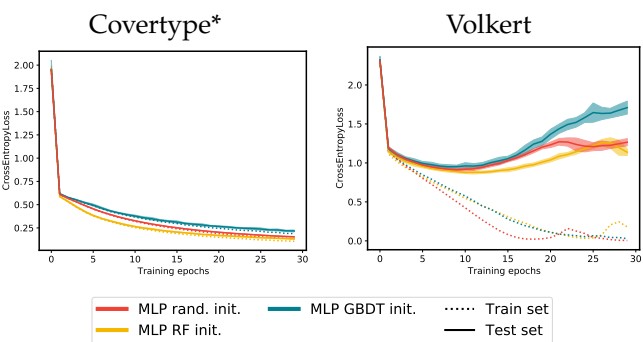

Figure 11: Optimization behaviour of randomly, RF and GBDT initialized MLP and SAINT evaluated over a 5 times repeated (stratified) 5-fold of each data set, according to Protocol P1, but where the MLP width is fixed to 2048 for all methods. The lines and shaded areas report the mean and standard deviation. *evaluation on a single 5-fold cross validation.

respect to a given method. The `optuna` library uses Bayesian optimization and, in particular, the tree-structured Parzen estimator model (Bergstra et al., 2011) to determine the parameters to be explored at each iteration of hyper-parameter optimization. This approach has been reported to outperform random search for hyper-parameter optimization (Turner et al., 2021).

**Data pre-processing** Machine learning pipelines often include pre-processing transformations of the input data before the training phase, a necessary step, especially when using neural networks (Bishop & from Geoffrey Hinton, 1995). We follow the pre-processing that is used in Borisov et al. (2021) and Somepalli et al. (2021). Hence, we normalize all continuous input features to zero mean and unit variance. This corresponds to linearly transform the input features as follows

$$\tilde{\mathbf{x}}_{:j} = \frac{\mathbf{x}_{:j} - \mu}{\sigma}$$

where $\mathbf{x}_{:j}$ is the $j$-th continuous feature of either train, validation or test observations, $\mu$ and $\sigma$ are the mean and standard deviation calculated over the train set only. This way we assure that no information from the validation or test sets is used in the normalization step. Moreover, all categorical features are label encoded, i.e. each level of a categorical variable is replaced with an integer in $\{1, \ldots, \#\,\text{levels}\}$.

### E.3 WORKING WITH AN ARBITRARY WIDTH IN P1 (OPTIMIZATION BEHAVIOUR)

Figure 11 shows the optimization behaviour of the randomly, RF and GBDT initialized MLP on the multi-class classification problems. Note that in contrast to Figure 2 in this setting, which is less restrictive for RF initialization, this method does indeed lead to *a faster convergence* and *a better minimum* (in terms of generalization).

However, for these multi-class classification problems, the GBDT initialization tends to deteriorate the optimization compared to RF or random initialization methods. Indeed, RF are genuinely multiclassification predictors whose splits are built using all output classes simultaneously whereas splits in GBDT are only built following a one-vs-all strategy. This implies that, with a fixed budget of splits (and therefore of neurons), RF are likely to be more versatile than GBDT.

### E.4 ADDITIONAL MATERIAL FOR PROTOCOL P2 (GENERALIZATION BEHAVIOUR)

### E.4.1 DETAILS ABOUT ADDITIONAL NN TRAINING TECHNIQUES

In Protocol 2, we assess the performances of generalization of the proposed methods, of the predictors described in Section 3.2, but also on three additional NN techniques:

1. the Xavier initialization (Glorot & Bengio, 2010) corresponds to a rescaled uniform initialization $U \sim \mathcal{U}\left[\pm \frac{\sqrt{6}}{\sqrt{n_{j+1}+n_j}}\right]$, where $n_j$ are the number of neurons in layer $j$. This random initialization is very close to the one used by default in this paper and simply denoted "random init";

2. the layer-sequential unit-variance orthogonal initialization (LSUV) (Mishkin & Matas, 2015) consists in a simple initialization that combines elements of (Glorot & Bengio, 2010) and (Saxe et al., 2013). In a first step, the weights of each layer are initialized as random orthogonal matrices. Then, the variance in the outputs on each layer on the training data is scaled close to 1 by repeatedly dividing the layer's weights by the empirically determined standard deviation. Although targeted to Computer Vision applications, this approach seems easily adaptable for our case;

3. the winning ticket network pruning (Frankle & Carbin, 2018) is more a simplification approach of the NN architecture during training than an initialization technique. That being said, it remains interesting to compare this strategy to the one developed in the paper, as the winning ticket network pruning enforces NN sparsity during training. This can be indeed put in parallel to the sparsity of the first layers introduced by the proposed initialization and preserved during training. The principle is to train a randomly initialized network, pruning it to obtain a sparse NN with similar performance and then re-train the sparse network a second time using the same instance of random initialization as before. These steps are repeated a certain number of times. The winning ticket network pruning is therefore computationally very intense and has to the best of our knowledge only been studied on medium-sized data sets. We thus use a slightly different procedure than (Frankle & Carbin, 2018) to determine winning tickets. First of all, we allocate at most $N$ training epochs to determining a winning ticket where $N$ is the number of epochs during the final model training itself. This fixed number of training epochs is then distributed among $n$ pruning rounds, each of which consists in training the model (for $N/n$ epochs), pruning it, and resetting all non-pruned weights to their initial (random) coefficients. This approach takes the same time as one-shot pruning but proves to be more efficient.

### E.4.2 EXTENSION OF TABLE 1 (BEST PERFORMANCES)

Table 4 provides a comparison of the performances obtained by ourselves and the literature (where available) for each model. Notice that our results are broadly consistent with those in the literature, with two exceptions. First, our random initialized MLP tends to perform better than in the literature, which can be explained by the fact that we use a much larger search space than usual for the MLP width (see Section 3.5 for a discussion on this). Second, our performance on Higgs is significantly lower than in the literature. This can be explained by the fact that we only include 5% of the original data set's observations in our analysis due to hardware limitations that do not allow us to train large MLP on 11M samples.

### E.4.3 BENEFITS OF TRAINING THE FEATURE EXTRACTOR VIA GRADIENT DESCENT

In Section 3, we demonstrated ways in which our initialization method can be beneficial for MLP training, resulting in faster convergence towards better minima (in the sens of generalization). A natural question that might arise in this context is whether translating the tree-based method into a MLP framework is actually beneficial. After all, one could be tempted to directly use the tree-based method as a feature pre-processing (without translating it into an MLP) and feed the resulting features into an MLP. In this case, the MLP would be trained via gradient descent *without* the feature extraction. However, it turns out that (i) the weights on the sparse feature extraction layer are indeed modified during gradient descent optimization and (ii) training the feature extractor via gradient descent largely contributes to the competitive generalization performance of our initialization method.

Figure 12 shows the histograms of the differences between all MLP parameters at initialization (RF strategy) and after training. As the histograms indicate, the weights in all layer

| Data set / Model | Housing (†) MSE ↓ | | Airbnb MSE ↓ ×10³ | Diamonds MSE ↓ ×10⁻³ | Covertype (†) Accuracy ↑ in % | | Volkert (§) Accuracy ↑ in % | |
| --- | --- | --- | --- | --- | --- | --- | --- | --- |
| | perf. in literature | our results | our results | our results | perf. in literature | our results | perf. in literature | our results |
| Random Forest | 0.272±0.006 | 0.263±0.009 | 5.39±0.13 | 9.80±0.35 | 78.1±0.1 | 83.6±0.1 | 66.3±1.3 | 64.2±0.3 |
| GBDT | 0.206±0.005 | **0.208±0.010** | **4.71±0.15** | **7.38±0.28** | 97.3±0.0 | **97.0±0.0** | 69.0±0.5 | 71.3±0.4 |
| Deep Forest | - | 0.225±0.008 | **4.68±0.16** | 8.23±0.29 | - | 92.4±0.1* | - | 66.3±0.4 |
| MLP rand. init. | 0.263±0.008 | 0.258±0.011 | 5.07±0.16 | 15.5±12.5 | 91.0±0.4 | 96.7±0.0 | 63.0±1.56 | 72.2±0.4 |
| MLP RF init. | - | 0.222±0.009 | **4.66±0.16** | 7.93±0.22 | - | 96.7±0.0 | - | **74.1±0.4** |
| MLP GBDT init. | - | **0.206±0.007** | 4.70±0.09 | 8.15±0.35 | - | 96.2±0.0 | - | 73.5±0.5 |
| MLP DF init. | - | 0.234±0.016 | **4.81±0.13** | 8.28±0.24 | - | 94.5±0.3* | - | 71.3±0.5 |
| SAINT | 0.226±0.004 | 0.258±0.011 | **4.81±0.15** | 17.7±3.83 | 96.3±0.1 | 96.6±0.1* | 70.1±0.6 | 70.1±0.4 |

| Data set / Model | Adult (†) AUC ↑ in % | | Bank (§) AUC ↑ in % | | Blastchar (§) AUC ↑ in % | | Heloc (†) AUC ↑ in % | | Higgs (†) AUC ↑ in % | |
| --- | --- | --- | --- | --- | --- | --- | --- | --- | --- | --- |
| | perf. in literature | our results | perf. in literature | our results | perf. in literature | our results | perf. in literature | our results | perf. in literature | our results |
| Random Forest | 91.7±0.2 | 91.6±0.3 | 89.1±0.3 | 92.8±0.3 | 80.6±0.7 | **84.5±1.2** | 90.0±0.2 | 91.3±0.6 | 79.7±0.0 | 80.4±0.1 |
| GBDT | 92.8±0.1 | **92.7±0.3** | 93.0±0.2 | **93.3±0.3** | 81.8±0.3 | **84.7±1.0** | 92.2±0.0 | **92.1±0.4** | 85.9±0.0 | 82.8±0.1 |
| Deep Forest | - | 91.8±0.3 | - | 92.9±0.2 | - | **83.7±1.2** | - | 90.3±0.5 | - | 81.2±0.0* |
| MLP rand. init. | 90.3±0.2 | 90.5±0.4 | 91.5±0.2 | 91.0±0.3 | 59.6±0.3 | 81.4±1.2 | 80.3±0.1 | 80.1±0.1 | 85.6±0.0 | 83.2±0.3 |
| MLP RF init | - | 92.1±0.3 | - | 92.4±0.4 | - | **84.4±1.2** | - | **91.7±0.4** | - | **83.6±0.1** |
| MLP GBDT init. | - | 92.2±0.3 | - | 92.5±0.3 | - | **84.6±1.2** | - | 91.5±0.6 | - | 83.0±0.0 |
| MLP DF init. | - | 91.9±0.4 | - | 92.2±0.3 | - | **84.2±1.0** | - | 91.4±0.6 | - | 83.3±0.1* |
| SAINT | 91.6±0.4 | 91.6±0.3 | 93.3±0.1 | 92.2±0.4 | 84.7±0.3 | 84.0±0.8 | 90.7±0.2 | 90.2±0.7 | 88.3±0.0 | **83.7±0.1*** |

Table 4: Best scores for Protocol P2. For each data set, our best overall score is highlighted in **bold** and our best Deep Learning score is underlined. Our scores are based on 5 times repeated (stratified) 5-fold cross validation. For each of our models, HP were selected via the `optuna` library (100 iterations). Sources for literature values: Borisov et al. (2021) (†) and Somepalli et al. (2021) (§). *score based on a single 5-fold cross validation.

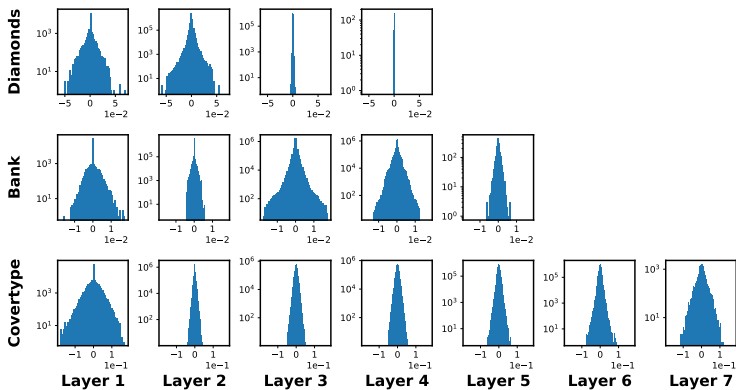

Figure 12: Histograms of the difference between all MLP parameters at initialization (RF strategy) and after training. Three data sets have been chosen for illustrative proposes. The behaviour in the light of our analysis (see E.4.3) is similar on the 7 other data sets.

throughout the MLP are modified during training. In particular, the weights of the first two (RF initialized) layers are not stationary but change to a large extent.

Table 5 shows the generalization performance of MLP initialized with the RF strategy, and compares the two scenarios in which the parameters of the first two layers (that is, the feature extraction layers built using the RF) are modified or frozen during MLP training. These results show that training the feature extraction layers is essential for the success of our initialization method.

| Data set
Model | Housing
MSE ↓ | Airbnb
MSE ↓
(x10³) | Diamonds
MSE ↓
(x10⁻³) | Adult
AUC ↑
(in %) | Bank
AUC ↑
(in %) | Blastchar
AUC ↑
(in %) | Heloc
AUC ↑
(in %) | Higgs
AUC ↑
(in %) | Covertype
Acc. ↑
(in %) | Volkert
Acc. ↑
(in %) |
|---|---|---|---|---|---|---|---|---|---|---|
| MLP rand. init. | 0.258±0.011 | 5.07±0.16 | 15.5±12.5 | 90.5±0.4 | 91.0±0.3 | 81.4±1.2 | 80.1±0.1 | 83.2±0.3 | 96.7±0.0 | 72.2±0.4 |
| MLP RF init. frozen | 0.262±0.018 | 14.5±2.71 | 13.7±1.48 | 91.1±0.3 | 90.9±0.5 | 84.4±0.9 | 91.0±0.6 | 75.9±0.2 | 92.2±0.2 | 69.4±0.5 |
| MLP RF init. | 0.222±0.009 | 4.66±0.16 | 7.93±0.22 | 92.1±0.3 | 92.4±0.4 | 84.4±1.2 | 91.7±0.4 | 83.6±0.1 | 96.7±0.0 | 74.1±0.4 |

Table 5: Best scores for Protocol 2. The scores are based on 5 times repeated (stratified) 5-fold cross validation. MLP RF init. frozen refers to the MLP RF init. model where the parameters of the first two layers (that are initialized using the Random Forest) are frozen during training, that is, they are kept at their initial values.

| Data set
Model | Housing | Airbnb | Diamonds | Adult | Bank | Blastchar | Heloc | Higgs | Covertype | Volkert |
|---|---|---|---|---|---|---|---|---|---|---|
| MLP rand. init. | 2.47M | 1.86M | 363k | 1.09M | 52.4K | 13.1M | 11.3M | 11.6M | 1.14M | 9.03M |
| MLP RF init. | 33.6M | 12.6M | 8.42M | 29.4M | 8.43M | 25.2M | 16.8M | 4.26M | 21.1M | 17.1M |
| MLP GBDT init. | 8.41M | 12.6M | 12.6M | 33.6M | 8.43M | 16.8M | 25.2M | 8.46M | 4.32M | 21.3M |
| MLP DF init. | 88.1M | 34.0M | 59.3M | 42.0M | 46.2M | 34.36M | 25.8M | 43.2M | 57.6M | 34.1M |
| SAINT | 56.8M | 27.0M | 53.1M | 7.20M | 6.12M | 322M | 98.2M | 43.2M | 6.44M | 169M |

Table 6: Comparison of the number of parameters for each model.

### E.4.4 NUMBER OF PARAMETERS OF BEST NEURAL NETWORKS

In Table 6, we compare the number of parameters of each NN method. Although the tree-based initialised MLP contain more parameters than the randomly initialized ones, the former are mostly sparse and the execution times are close (see Table 7). Finally note that the number of parameters of the RF/GBDT init. MLP is globally on par with that of SAINT (sometimes more, sometimes less) but for a smaller execution times (Table 7) and mostly better performances (Table 4).

### E.4.5 COMPARISON OF THE EXECUTION TIMES OF THE BEST NEURAL NETWORKS

Table 7 presents a comparison of the execution times of the training of different NN methods using the hyper-parameters determined by the protocol P2. For each model, the total training time (initialization + gradient descent optimization) is given, measured up to the point where the best validation loss is reached ("early stopping"). It shows that RF/GBDT initialized MLP train faster than SAINT and a bit slower than randomly initialized MLP. For completeness, Table 8 gives the execution time for the initialization and training step separately.

### E.4.6 OPTIMIZATION BEHAVIOUR

For completeness, Figure 13 shows the optimization behaviour of the randomly, RF and GBDT initialized MLP as well as SAINT under the Protocol P2.

| Data set
Model | Housing | Airbnb | Diamonds | Adult | Bank | Blastchar | Heloc | Higgs | Covertype | Volkert |
|---|---|---|---|---|---|---|---|---|---|---|
| MLP rand. init. | 5.96 (32) | 91.9 (98) | 13.3 (31) | 11.8 (37) | 21.6 (62) | 6.78 (34) | 14.3 (60) | 467 (32) | 312 (69) | 12.3 (31) |
| MLP RF init. | 37.8 (29) | 131 (44) | 26.8 (25) | 17.2 (19) | 21.5 (23) | 8.58 (18) | 6.61 (15) | 253 (39) | 2040 (91) | 28.3 (25) |
| MLP GBDT init. | 22.9 (49) | 280 (95) | 53.6 (37) | 34.5 (31) | 7.47 (3) | 7.76 (7) | 8.54 (8) | 63.0 (5) | 437 (66) | 52.8 (37) |
| MLP DF init. | 233 (72) | 360 (48) | 182 (31) | 99.4 (54) | 105 (26) | 29.0 (52) | 14.7 (19) | 3280 (76) | 5580 (95) | 181 (31) |
| SAINT | 81.9 (37) | 640 (83) | 394 (84) | 15.6 (11) | 52.7 (32) | 7.23 (2) | 51.0 (31) | 2310 (19) | 6580 (97) | 394 (84) |

Table 7: Comparison of the execution time in seconds for model initialization and training until the best validation lost is reached. The number of training epochs is indicated in parentheses.

| Data set
Model | Housing | Airbnb | Diamonds | Adult | Bank | Blastchar | Heloc | Higgs | Covertype | Volkert |
|---|---|---|---|---|---|---|---|---|---|---|
| MLP rand. init. | 0.00/5.96 (32) | 0.00/91.9 (98) | 0.00/13.3 (31) | 0.00/11.8 (37) | 0.00/21.6 (62) | 0.00/6.78 (34) | 0.00/14.3 (60) | 0.00/467 (32) | 0.00/312 (69) | 0.00/12.3 (31) |
| MLP RF init. | 2.21/35.6 (29) | 1.87/129 (44) | 2.20/24.6 (25) | 1.16/16.0 (19) | 1.89/19.6 (23) | 2.81/5.77 (18) | 1.78/4.83 (15) | 6.31/247 (39) | 3.67/2040 (91) | 3.70/24.6 (25) |
| MLP GBDT init. | 5.35/17.5 (49) | 4.19/276 (95) | 4.65/48.9 (37) | 2.32/32.2 (31) | 4.27/3.20 (3) | 6.07/1.69 (7) | 4.65/3.89 (8) | 4.18/58.8 (5) | 2.30/435 (66) | 3.88/48.9 (37) |
| MLP DF init. | 15.1/218 (72) | 5.31/355 (48) | 7.19/175 (31) | 8.36/91 (54) | 9.25/96 (26) | 6.64/22.3 (52) | 5.64/9.04 (19) | 18.9/3260 (76) | 11.2/5570 (95) | 5.87/175 (31) |

Table 8: Comparison of the execution time in seconds for MLP initialization/training until the best validation lost is reached. The number of training epochs is indicated in parentheses. A value of $0.00$ indicates running times smaller than $5 \times 10^{-3}$ seconds.

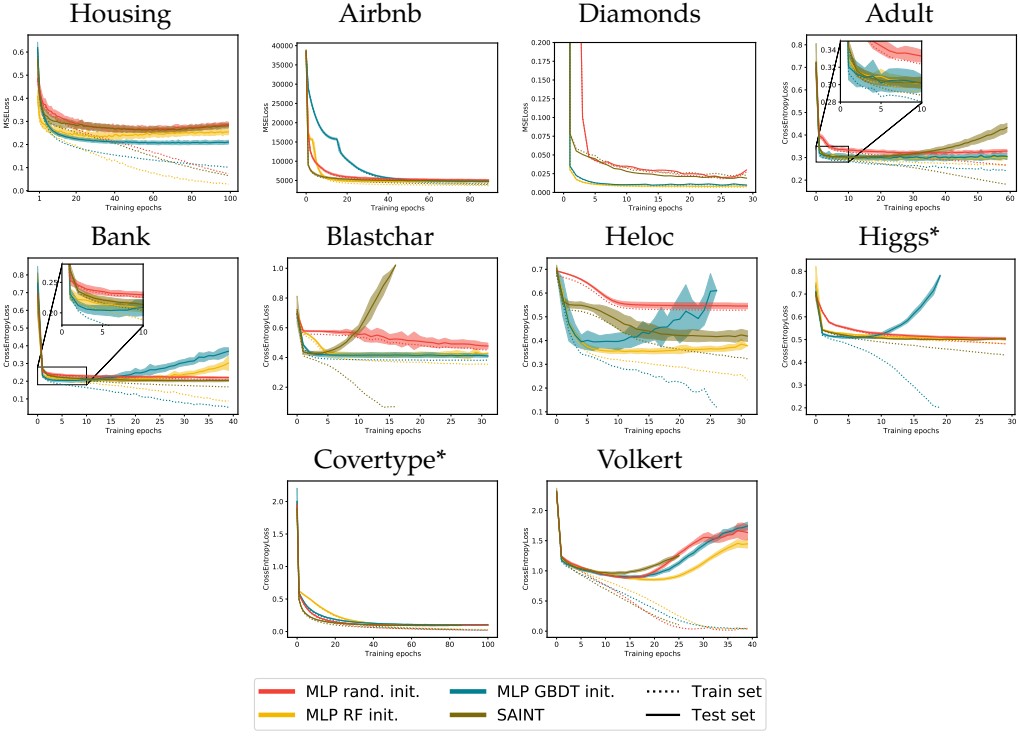

Figure 13: Optimization behaviour of randomly, RF and GBDT initialized MLP and SAINT evaluated over a 5 times repeated (statisfied) 5-fold of each data set, according to Protocol P2. The lines and shaded areas report the mean and standard deviation. *evaluation on a single 5-fold cross validation.

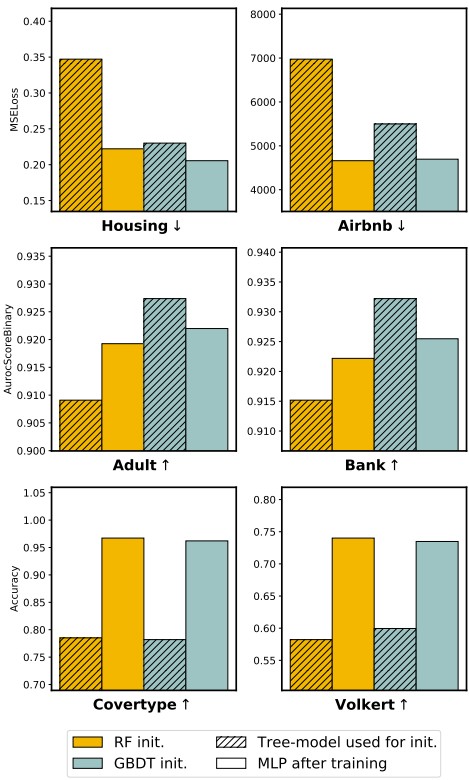

Figure 14: Comparison of the performance of the RF and GBDT models used for initialization and the final performance of the corresponding MLPs.

### E.5    Hyper-parameter setting

#### E.5.1    Search spaces

Table 9 shows the HP search spaces that were used to determine an optimal HP setting. The same search spaces were used for the experimental protocols P1 and P2. Note that, in Table 9, n_classes corresponds to the number of classes for classification problems and is 1 for regression problems. Furthermore, the different search spaces given for SAINT were used for smaller/larger data sets, where a data set qualifies as smaller if it has less that 50 explanatory variables.

#### E.5.2    Experimental protocol P2

Tables 10 and 11 show the HP setting used for the experimental protocol P2. For the search spaces and descriptions of the function of each HP see Table 9.

### E.6    Performances of tree-based methods used for initialisation of MLP

Figure 14 compares the performance of RF and GBDT models and the performance of optimized MLP, initialized with RF and GBDT respectively. We can notice that the difference in performance between GBDT and RF does not systematically turn into the same difference in performance for the corresponding trained networks. This suggests that beyond their respective performances, the very structures of RF and GBDT predictors play an important role in the final MLP performances.

| Method | Parameter | Search space | Function |
|---|---|---|---|
| Random Forests | max_depth | $\{1, \ldots, 12\}$ | |
| | n_estimators | $\{1000\}$ | see here |
| | max_features | $[0, 1]$ | |
| GBDT | max_depth | $\{1,\ldots,12\}$ | |
| | n_estimators | $\{1000\}$ | |
| | reg_alpha | $[10^{-8}, 1]$ | see here |
| | reg_lambda | $[10^{-8}, 1]$ | |
| | learning_rate | $[0.01, 0.3]$ | |
| Deep Forest | forest_depth | $\{1, 2, 3\}$ | Number of Deep Forest layers |
| | n_forests | $\{1\}$ | Number of forests per Deep Forest layer |
| | n_estimators | $\{1000\}$ | |
| | max_depth | $\{1, \ldots, 12\}$ | RF parameters, see here |
| | max_features | $[0, 1]$ | |
| MLP random init. | learning_rate | $[10^{-6}, 10^{-1}]$ | learning rate of SGD training |
| | depth | $\{1, \ldots, 10\}$ | number of layer |
| | width | $\{1, \ldots, 2048\}$ | number of neurons per layer |
| | epochs | $\{100\}$ | number of SGD training epochs |
| | batch_size | $\{256\}$ | batch size of SGD training |
| MLP RF init. | max_depth | $\{1, \ldots, 11\}$ | |
| | n_estimators | $2048/2^{\text{max\_depth}}$ | Parameters of the RF initializer, see here |
| | max_features | $[0, 1]$ | |
| | learning_rate | $[10^{-6}, 10^{-1}]$ | learning rate of SGD training |
| | depth | $\{3, \ldots, 10\}$ | number of layer |
| | width | $\{2048\}$ | number of neurons per layer |
| | epochs | $\{100\}$ | number of SGD training epochs |
| | batch_size | $\{256\}$ | batch size of SGD training |
| | strength01 | $[1, 10^4]$ | |
| | strength12 | $[0.01, 100]$ | MLP translation parameters, see Section 2.3 |
| MLP GBDT init. | max_depth | $\{1,\ldots,11\}$ | |
| | n_estimators | $2048/(\text{n\_classes} \cdot 2^{\text{max\_depth}})$ | |
| | reg_alpha | $[10^{-8}, 1]$ | Parameters of the GBDT initializer, see here |
| | reg_lambda | $[10^{-8}, 1]$ | |
| | learning_rate_GBDT | $[0.01, 0.3]$ | |
| | learning_rate | $[10^{-6}, 10^{-1}]$ | learning rate of SGD training |
| | depth | $\{3, \ldots, 10\}$ | number of layer |
| | width | $\{2048\}$ | number of neurons per layer |
| | epochs | $\{100\}$ | number of SGD training epochs |
| | batch_size | $\{256\}$ | batch size of SGD training |
| | strength01 | $[1, 10^4]$ | |
| | strength12 | $[0.01, 100]$ | MLP translation parameters, see Section 2.3 |
| MLP DF init. | forest_depth | $\{1, 2, 3\}$ | Number of Deep Forest layers |
| | n_forests | $\{1\}$ | Number of forests per Deep Forest layer |
| | n_estimators | $2048/2^{\text{max\_depth}}$ | |
| | max_depth | $\{1, \ldots, 12\}$ | RF parameters, see here |
| | max_features | $[0, 1]$ | |
| | learning_rate | $[10^{-6}, 10^{-1}]$ | learning rate of SGD training |
| | depth | $\{3, \ldots, 10\}$ | number of layer |
| | width | $\{2048\}$ | number of neurons per layer |
| | epochs | $\{100\}$ | number of SGD training epochs |
| | batch_size | $\{256\}$ | batch size of SGD training |
| | strength01 | $[1, 10^4]$ | |
| | strength12 | $[0.01, 100]$ | |
| | strength23 | $[0.01, 100]$ | MLP translation parameters, see Section 2.3 |
| | strength_id | $[0.01, 100]$ | |
| SAINT | epochs | $\{100\}$ | number of SGD training epochs |
| | batch_size | $\{256\}/\{64\}$ | batch size of SGD training |
| | dim | $[32, 64, 128]/[8, 16]$ | number of neurons per layer in attention block |
| | depth | $\{1, 2, 3\}$ | number of layers in each attention block |
| | heads | $\{2, 4, 8\}$ | number of head in each attention layer |
| | dropout | $\{0, 0.1, 0.2, 0.3, 0.4, 0.5, 0.6, 0.7, 0.8\}$ | dropout used during SGD training |

Table 9: Hyper-parameter search spaces used for numerical evaluations.

| Method | Parameter | Housing | Airbnb | Adult | Bank | Covertype | Volkert |
|---|---|---|---|---|---|---|---|
| Random Forests | max_depth | 12 | 12 | 11 | 12 | 12 | 12 |
| | n_estimators | 1000 | 1000 | 1000 | 1000 | 1000 | 1000 |
| | max_features | 0.437 | 0.623 | 0.596 | 0.943 | 0.811 | 0.688 |
| GBDT | max_depth | 12 | 9 | 6 | 7 | 11 | 10 |
| | n_estimators | 1000 | 1000 | 1000 | 1000 | 1000 | 1000 |
| | reg_alpha | 0.305 | $4.60 \times 10^{-6}$ | $2.39 \times 10^{-5}$ | $1.52 \times 10^{-4}$ | 0.728 | $4.47 \times 10^{-6}$ |
| | reg_lambda | $1.13 \times 10^{-2}$ | $1.75 \times 10^{-8}$ | $1.35 \times 10^{-6}$ | $1.07 \times 10^{-3}$ | $6.51 \times 10^{-4}$ | $1.71 \times 10^{-6}$ |
| | learning_rate | $3.82 \times 10^{-2}$ | 0.238 | $1.08 \times 10^{-2}$ | $1.34 \times 10^{-2}$ | 0.181 | 0.107 |
| Deep Forest | forest_depth | 4 | 9 | 2 | 2 | 9 | 3 |
| | n_forests | 1 | 1 | 1 | 1 | 1 | 1 |
| | n_estimators | 1000 | 1000 | 1000 | 1000 | 1000 | 1000 |
| | max_depth | 5 | 12 | 11 | 9 | 12 | 12 |
| | max_features | 0.361 | 0.410 | 0.166 | 0.206 | 0.218 | 0.134 |
| MLP random init. | learning_rate | $9.01 \times 10^{-4}$ | $4.21 \times 10^{-4}$ | $2.07 \times 10^{-4}$ | $1.1 \times 10^{-4}$ | $1.15 \times 10^{-4}$ | $2.29 \times 10^{-4}$ |
| | depth | 4 | 4 | 4 | 4 | 4 | 6 |
| | width | 1100 | 959 | 1175 | 856 | 738 | 1482 |
| | epochs | 100 | 100 | 100 | 100 | 100 | 100 |
| | batch_size | 256 | 256 | 256 | 256 | 256 | 256 |
| MLP RF init. | max_depth | 8 | 10 | 8 | 8 | 10 | 8 |
| | n_estimators | 8 | 2 | 8 | 8 | 2 | 8 |
| | max_features | 0.442 | 0.321 | 0.613 | 0.650 | 0.897 | 0.825 |
| | learning_rate | $1.04 \times 10^{-4}$ | $1.72 \times 10^{-4}$ | $1.55 \times 10^{-5}$ | $1.01 \times 10^{-4}$ | $1.04 \times 10^{-5}$ | $1.45 \times 10^{-4}$ |
| | depth | 10 | 5 | 5 | 4 | 7 | 6 |
| | width | 2048 | 2048 | 2048 | 2048 | 2048 | 2048 |
| | epochs | 100 | 100 | 100 | 100 | 100 | 100 |
| | batch_size | 256 | 256 | 256 | 256 | 256 | 256 |
| | strength01 | 1090 | 668 | 537 | 71.4 | 13.7 | 1.02 |
| | strength12 | 0.0749 | 1.09 | 62.7 | 34.5 | $1.05 \times 10^{-2}$ | $5.53 \times 10^{-2}$ |
| MLP GBDT init. | max_depth | 3 | 4 | 4 | 4 | 8 | 4 |
| | n_estimators | 256 | 128 | 128 | 128 | 1 | 12 |
| | reg_alpha | $1.30 \times 10^{-7}$ | $1.10 \times 10^{-2}$ | $1.26 \times 10^{-8}$ | 0.413 | $1.33 \times 10^{-2}$ | $6.76 \times 10^{-6}$ |
| | reg_lambda | $1.57 \times 10^{-7}$ | $9.52 \times 10^{-4}$ | $7.85 \times 10^{-4}$ | $7.48 \times 10^{-3}$ | 0.643 | $1.99 \times 10^{-7}$ |
| | learning_rate_GBDT | 0.211 | 0.297 | 0.202 | 0.285 | 0.112 | 0.272 |
| | learning_rate | $1.11 \times 10^{-5}$ | $1.97 \times 10^{-5}$ | $4.77 \times 10^{-5}$ | $6.22 \times 10^{-4}$ | $6.19 \times 10^{-5}$ | $1.63 \times 10^{-4}$ |
| | depth | 4 | 5 | 6 | 4 | 3 | 7 |
| | width | 2048 | 2048 | 2048 | 2048 | 2048 | 2048 |
| | epochs | 100 | 100 | 100 | 100 | 100 | 100 |
| | batch_size | 256 | 256 | 256 | 256 | 256 | 256 |
| | strength01 | 575 | 7830 | 132 | 20.5 | 7280 | 4.08 |
| | strength12 | 5.60 | 0.461 | 66.0 | 5.52 | 93.4 | $7.11 \times 10^{-2}$ |
| MLP DF init. | forest_depth | 6 | 3 | 3 | 2 | 3 | 2 |
| | n_forests | 1 | 1 | 1 | 1 | 1 | 1 |
| | n_estimators | 16 | 2 | 64 | 32 | 2 | 8 |
| | max_depth | 7 | 10 | 5 | 6 | 10 | 8 |
| | max_features | 0.350 | 0.598 | 0.992 | 0.322 | 0.633 | 0.342 |
| | learning_rate | $1.04 \times 10^{-5}$ | $6.67 \times 10^{-5}$ | $1.54 \times 10^{-5}$ | $3.08 \times 10^{-5}$ | $1.58 \times 10^{-5}$ | $2.31 \times 10^{-4}$ |
| | depth | 23 | 10 | 9 | 13 | 15 | 9 |
| | width | 2048 | 2048 | 2048 | 2048 | 2048 | 2048 |
| | epochs | 100 | 100 | 100 | 100 | 100 | 100 |
| | batch_size | 256 | 256 | 256 | 256 | 256 | 256 |
| | strength01 | 515 | 36.6 | 41.0 | 15.5 | 51.6 | 1.41 |
| | strength12 | 0.162 | 0.242 | 10.6 | 0.213 | 0.124 | 0.154 |
| | strength23 | 1.94 | 10.4 | 47.8 | 1.94 | $4.26 \times 10^{-2}$ | 0.149 |
| | strength_id | $3.63 \times 10^{-2}$ | $6.34 \times 10^{-2}$ | 7.44 | $2.75 \times 10^{-2}$ | $5.09 \times 10^{-2}$ | 3.69 |
| SAINT | epochs | 100 | 100 | 100 | 100 | 100 | 100 |
| | batch_size | 256 | 256 | 256 | 256 | 64 | 256 |
| | dim | 128 | 64 | 32 | 32 | 8 | 16 |
| | depth | 3 | 2 | 2 | 1 | 2 | 2 |
| | heads | 2 | 8 | 2 | 8 | 4 | 8 |
| | dropout | 0.2 | 0 | 0.4 | 0.8 | 0.5 | 0.8 |

Table 10: Hyper-parameters used for the experimental protocol P2.

| Method | Parameter | Diamonds | Blastchar | Heloc | Higgs |
|---|---|---|---|---|---|
| Random Forests | max_depth | 12 | 6 | 9 | 12 |
| | n_estimators | 1000 | 1000 | 1000 | 1000 |
| | max_features | 0.967 | 0.547 | 0.607 | 0.577 |
| GBDT | max_depth | 7 | 1 | 1 | 11 |
| | n_estimators | 1000 | 1000 | 1000 | 1000 |
| | reg_alpha | 0.341 | $7.15 \times 10^{-7}$ | 0.123 | $2.29 \times 10^{-8}$ |
| | reg_lambda | $5.15 \times 10^{-4}$ | $1.59 \times 10^{-7}$ | $1.44 \times 10^{-2}$ | 0.391 |
| | learning_rate | $9.17 \times 10^{-2}$ | $1.48 \times 10^{-2}$ | 0.282 | $2.46 \times 10^{-2}$ |
| Deep Forest | forest_depth | 4 | 7 | 10 | 3 |
| | n_forests | 1 | 1 | 1 | 1 |
| | n_estimators | 1000 | 1000 | 1000 | 1000 |
| | max_depth | 12 | 2 | 4 | 12 |
| | max_features | 0.454 | 0.641 | 0.196 | 0.163 |
| MLP random init. | learning_rate | $2.35 \times 10^{-4}$ | $1.05 \times 10^{-4}$ | $1.14 \times 10^{-6}$ | $2.26 \times 10^{-5}$ |
| | depth | 9 | 8 | 8 | 9 |
| | width | 1011 | 1475 | 1369 | 1284 |
| | epochs | 100 | 100 | 100 | 100 |
| | batch_size | 256 | 256 | 256 | 256 |
| MLP RF init. | max_depth | 10 | 5 | 7 | 9 |
| | n_estimators | 2 | 64 | 16 | 4 |
| | max_features | 0.904 | 0.425 | 0.728 | 0.670 |
| | learning_rate | $6.67 \times 10^{-5}$ | $5.07 \times 10^{-6}$ | $7.33 \times 10^{-6}$ | $2.17 \times 10^{-5}$ |
| | depth | 4 | 8 | 6 | 3 |
| | width | 2048 | 2048 | 2048 | 2048 |
| | epochs | 100 | 100 | 100 | 100 |
| | batch_size | 256 | 256 | 256 | 256 |
| | strength01 | 19.8 | 4500 | 331 | 1.43 |
| | strength12 | 0.420 | 42.9 | 1.06 | 0.329 |
| MLP GBDT init. | max_depth | 3 | 1 | 3 | 5 |
| | n_estimators | 256 | 1024 | 256 | 64 |
| | reg_alpha | $4.56 \times 10^{-2}$ | $1.63 \times 10^{-5}$ | $6.21 \times 10^{-7}$ | $2.58 \times 10^{-6}$ |
| | reg_lambda | $6.17 \times 10^{-4}$ | $2.19 \times 10^{-4}$ | $3.03 \times 10^{-4}$ | $3.20 \times 10^{-6}$ |
| | learning_rate_GBDT | 0.214 | $4.72 \times 10^{-2}$ | $8.42 \times 10^{-2}$ | 0.290 |
| | learning_rate | $8.94 \times 10^{-5}$ | $5.60 \times 10^{-6}$ | $4.54 \times 10^{-4}$ | $1.36 \times 10^{-4}$ |
| | depth | 5 | 6 | 8 | 4 |
| | width | 2048 | 2048 | 2048 | 2048 |
| | epochs | 100 | 100 | 100 | 100 |
| | batch_size | 256 | 256 | 256 | 256 |
| | strength01 | 3870 | 4690 | 6550 | 4780 |
| | strength12 | 56.6 | 21.0 | 31.8 | 0.423 |
| MLP DF init. | forest_depth | 3 | 2 | 2 | 3 |
| | n_forests | 1 | 1 | 1 | 1 |
| | n_estimators | 4 | 128 | 64 | 8 |
| | max_depth | 9 | 4 | 5 | 8 |
| | max_features | 0.695 | 0.516 | 0.280 | 0.572 |
| | learning_rate | $2.04 \times 10^{-5}$ | $2.00 \times 10^{-6}$ | $1.91 \times 10^{-5}$ | $9.33 \times 10^{-6}$ |
| | depth | 16 | 10 | 8 | 12 |
| | width | 2048 | 2048 | 2048 | 2048 |
| | epochs | 100 | 100 | 100 | 100 |
| | batch_size | 256 | 256 | 256 | 256 |
| | strength01 | 21.0 | 93.0 | 97.8 | 1.12 |
| | strength12 | 0.119 | 20.0 | 0.987 | $9.22 \times 10^{-2}$ |
| | strength23 | $5.34 \times 10^{-2}$ | 0.283 | 27.1 | 0.207 |
| | strength_id | 0.358 | 0.475 | 9.70 | 0.152 |
| SAINT | epochs | 100 | 100 | 100 | 100 |
| | batch_size | 256 | 256 | 256 | 64 |
| | dim | 64 | 128 | 64 | 16 |
| | depth | 3 | 3 | 3 | 2 |
| | heads | 4 | 8 | 2 | 8 |
| | dropout | 0.2 | 0.5 | 0.8 | 0.8 |

Table 11: Hyper-parameters used for the experimental protocol P2.

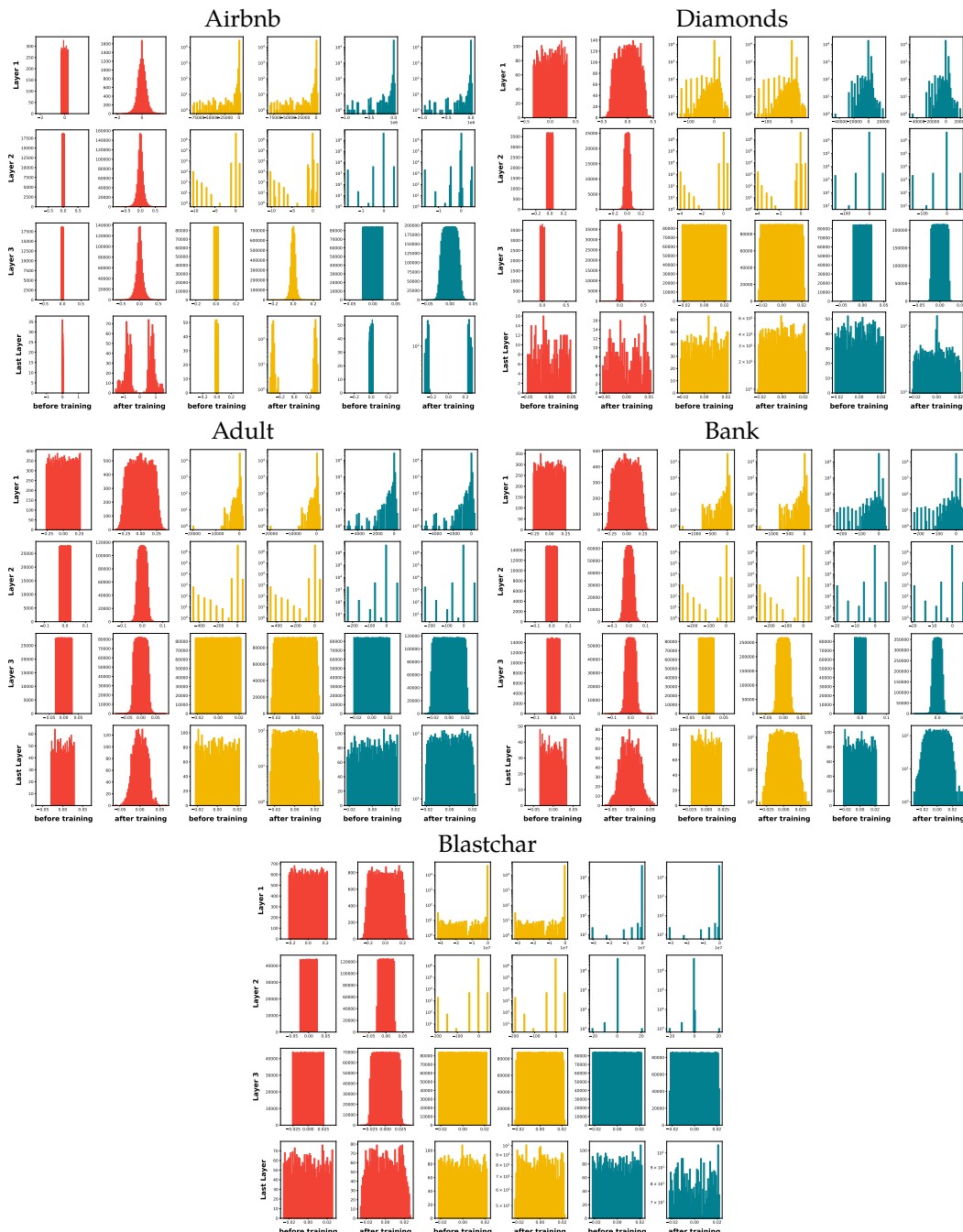

Figure 15: Histograms of the first three first and the last layers' weights before and after the MLP training on the Airbnb, Diamonds, Adult, Bank and Blastchar data sets. Comparison between random, RF and GBDT initializations.

### E.7 ADDITIONAL FIGURES TO SECTION 3.5 (ANALYZING KEY ELEMENTS OF THE NEW INITIALIZATION METHODS)

Figures 15 and 16 show the same histograms as Figure 4 evaluated on the other data set considered in protocol P2. Note the logarithmic y-axis for the first two RF and GBDT initialized layers.

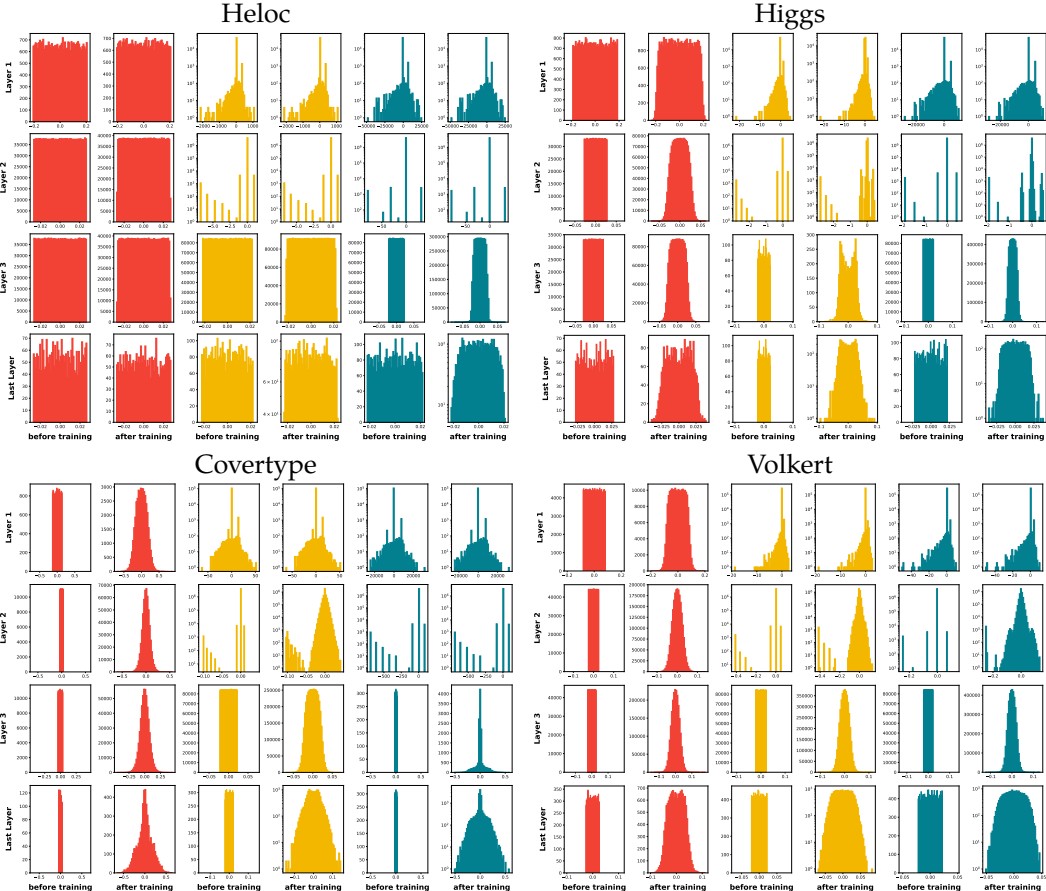

Figure 16: Histograms of the first three first and the last layers' weights before and after the MLP training on the Heloc, Higgs, Covertype and Volkert data sets. Comparison between random, RF and GBDT initializations.

