# OpenReview forum: "Sparse tree-based Initialization for Neural Networks"
_ICLR.cc/2023/Conference — ICLR 2023 poster_

### Official Review · Reviewer_mdb9 · 2022-10-24

**Confidence:** 3
**Correctness:** 3
**Technical Novelty And Significance:** 3
**Empirical Novelty And Significance:** 3
**Recommendation:** 8

**Clarity, Quality, Novelty And Reproducibility:**

These are some suggestions that I'd like make to the authors in order to further improve their draft:

- In Related Works section, please make emphasis and/or describe briefly how your approach herein proposed for transforming a DT into a MLP differentiates from methods previously found in literature. In this section you mention that these _translations_ have been proposed as early as the 90s; since this is such a critical part of your approach (and also because is a fundamental good writing style) I think you should make clear as soon as possible how this new translating mechanism differentiates from other approaches previously proposed. Otherwise, novelty might be put into question.

- It would also be good if you discuss some of the motivations that led to your research. From the wording in the abstract and intro, I got the feeling that you consider neural nets shall "reign supreme" overall other ML methods, and if they lose against any other ML paradigm in certain areas of application, then we shall bring elements of such other family of methods to deep learning, until they are "the one ML method to rule them all"... that'd be a weird line of thought. Think of the following two questions => "Why cannot NN be as good, or even better, than tree methods in this or that kind of problems?" vs "How can we enhance the performance of NN by taking elements of other ML paradigms?" which one sounds more natural to you? Just a bit of rephrasing in abstract and intro would do the trick in my opinion.

- Did you consider discussing the implications of your research regarding the capabilities and limitations of gradient methods and the overall optimization complexity of neural nets? I think you might be sitting on something beyond just an initialization technique, but rather further evidence that gd is not as good as previously thought, which has been a recent line of research (e.g. lottery ticket hypothesis, etc.).

**Strength And Weaknesses:**

- The paper is very well written in structure, language and style, with only minor issues here and there.

- The proposed method is very interesting, although remarkably simple. Authors do a very good job explaining it, nonetheless.

- Experimental evidence provided is comprehensive, and provides good empirical support for the proposed approach.

- Results are highly encouraging to continue this line of research, where different ML algorithms combine and blend, in simple and very effective ways.

- Authors also do not fall short in providing discussion regarding all aspects of their works (inner workings and behavior of their methods, results, consequences and possibilities resulting from their  work).

**Summary Of The Paper:**

Authors propose a method to 'translate' a Decision Tree (DT) into a Multi layer perceptron (MLP). They use this method as the basis for a Neural Net (NN) weights' initialization technique that allows them to considerably boost NN's performance on tabular (unstructured) data. Authors provide a thorough set of experimental tests where they compare the performances of their proposed approach to that of standard (random) NN init, as well as against the performances of different DT-based methods and another NN architecture aimed at  tabular data. The results obtained back the validity of their method.

**Summary Of The Review:**

In general, I think this paper is very good. My only doubt is how similar is this DT-MLP transformation technique to approaches previously proposed in the field. Other than that, I think the paper is well written and presented, experimental evidence and results check, and novelty and significance are good enough for ICLR.

---

> ### Author Response · Authors · 2022-11-18
> **Specific comments**
>
>
> **Position of the work** Please refer to the general comment.
>
> **Rephrasing the motivation** We agree with the point raised by the reviewer, and we have reformulated the motivations to underline how NN predictors could benefit from tree-based techniques. Please see the revised version for modifications.
>
>
> **About GD efficiency in NN**
> A recent line of research has focused on the properties of GD-based training for specific weight distributions, specific neural network architectures and over-parametrized regimes.
> More precisely, Woodworth et al. (2020) and Pesme et al. (2021)
> prove, in the case of a diagonal (therefore linear) NN, that SGD optimization can operate an implicit $\ell^1$ or $\ell^2$ regularization depending on the variance of the distribution of the initial weights.
> This paradigm can be hardly applied to practical neural network with non-linear activation functions, trained with a finite number of observations, and not excessively over-parameterized.
> We adopt a different point of view in our work by designing a sparse tree-based initialization for the first layers of our neural networks. It turns out that the SGD-like optimization procedure manages to keep the weight sparsity through training, leading to a final predictor with improved generalization properties, compared to (dense) uniform initialization.
> Therefore, our work highlights that SGD optimizer fails to find the best final weights starting from a random and unstructured initialization. Hence, the implicit regularization that operates with SGD in a specific theoretical framework does not seem strong enough in practice to lead to a good optimum, without a specific (sparse) weight initialization.
> The fact that the weight sparsity is preserved through training could potentially result from a more complex SGD regularization, which will be definitely interesting to study in future works.

---

### Official Review · Reviewer_znja · 2022-10-25

**Confidence:** 4
**Correctness:** 3
**Technical Novelty And Significance:** 3
**Empirical Novelty And Significance:** 3
**Recommendation:** 6

**Clarity, Quality, Novelty And Reproducibility:**

The paper is clear and logically structured. Unfortunately, the code is unavailable, making it difficult to reproduce the experiments.
The idea of using tree-based methods to initialize NN models is simple but seems effective considering the paper's results. However, the idea is not entirely new, and the authors should compare the proposed model with other approaches and with other initialization methods. Therefore, the comparison with random weight initialization seems not fair.


**Details Of Ethics Concerns:**

I have no ethical concerns about the paper.

**Strength And Weaknesses:**

+ The proposed approach is interesting and presents promising results.
+ The idea of using decision trees to capture relevant features and their interactions and define a mapping to encode extracted relationships into a neural network is not new. Still, the authors presented a focused approach and interesting experiments to justify its use.

- The paper suggests the proposed approach operates in an implicit regularization during the NN training. However, I’m not convinced that it was entirely demonstrated in the experiments, results, and analysis.
- The authors compared the proposed NN initialization approach with random weights initialization. However, this is an active research area, and classical approaches should be considered for comparison:
   - Understanding the difficulty of training deep feedforward neural networks, Glorot and Bengio, 2010
   - Exact solutions to the nonlinear dynamics of learning in deep linear neural networks, Saxe et al., 2013
   - Random walk initialization for training very deep feedforward networks, Sussillo and Abbott, 2014
   - Data-dependent Initializations of Convolutional Neural Networks, Krähenbühl et al., 2015
   - All you need is a good init, Mishkin and Matas, 2015
   - Fixup Initialization: Residual Learning Without Normalization, Zhang et al., 2019
   - The Lottery Ticket Hypothesis: Finding Sparse, Trainable Neural Networks, Frankle and Carbin, 2019
- The authors do not consider the computational cost of training the tree-based methods to initialize the NN model.
- What is the performance of the proposed approach on deeper models or using different activation functions? Please provide more comments on that.
- The authors did not discuss the limitation of the proposed method. Therefore, it will be meaningful to discuss the gap between the experiments in the current version of the paper and the real-world applications.


**Summary Of The Paper:**

This paper proposes a method to initialize MLP neural networks when working with tabular data. The method uses tree-based techniques to initialize the weights of the NN. The authors argue that using this initialization in the first layer of an MLP is sufficient to improve its performance (the initialization of the other layers is random).

**Summary Of The Review:**

The proposed paper presents a promising approach with interesting results and analysis. The authors focus on comparing the proposed method with random initialization methods but should consider other initialization approaches to compare. The author also should discuss the proposed approach's limitations.

---

> ### Author Response · Authors · 2022-11-18
> **Specific comments**
>
> **Link with implicit regularisation** In the submitted paper, we use the term ``implicit regularization" to refer to the fact that the optimization strategy employed during training preserves the weight structure injected at initialization.
> We have clarified this point in the paper.
>
>
> **Random initialisation as a baseline** The random initialization used in our numerical experiments is one of the most commonly used and is the one chosen by default in the Pytorch library.
> Please refer to the general comment for a discussion about other newly implemented initialization techniques for comparison.
>
>
> **Computational cost of the method** The reviewer is right. The training times given only account for the MLP training, which is usually *vastly* more time-consuming than the random or tree-based initialization step (tree-based methods usually being much faster to train than neural networks). We have added the time costs of initialization vs. training for completeness, see Table 8 in the revised version. The ratio between initialization times and training times remains in favor of the proposed initialization technique as the latter allows to improve significantly the performances of the resulting NN, and is still faster than using more complex NN architectures such as transformers.
>
>
> **Performances of deeper NN** The HP search range for MLP depth is $\{1,\dots,10\}$. 10 layers can already be considered deep in most cases as few data are available. In all our numerical experiments, the maximum depth of 10 was only chosen by a single model (the Housing dataset when using a RF init). In this last case, the tree-based initialization still improves over the standard random one.
>
> **Choice of activation function** A priori, simply using a ReLU instead of the carefully calibrated tanh function should degrade the quality of the tree translation which would in turn diminish the benefits drawn from the translation by the following layers. However, it might be interesting to try tuning ReLU (or a close version of ReLU) such that it approximates the indicator function and observe the performances (optimisation and generalisation) of the resulting network. Due to time constraints, we had no time to implement such methods, but this would be definitely a promising venue for future work.
>
>
> **Gap between XP and real world applications** All our evaluations are conducted on heterogeneous, real-world data sets as it is a standard assessment typically done in the ML community. An application to a specific real-life case to solve a concrete problem is of high interest, but remains beyond the scope of this work, as a whole complementary line of research.
>
> **Discussion on limitations** You are right, the limitations of our work were disseminated across the document. For more clarity, as you suggested, we have added a paragraph in the conclusion of the revised version in order to explicitly discuss the limitations of the proposed method.
>
>
> **Code accessibility** For reproducibility purpose, the code for all numerical experiments has been made public on an anonymous Git repository https://github.com/gbW3w/Sparse_Tree-Based_Initialization_for_NN/tree/master.

---

### Official Review · Reviewer_c3Rb · 2022-10-26

**Confidence:** 2
**Correctness:** 3
**Technical Novelty And Significance:** 1
**Empirical Novelty And Significance:** 1
**Recommendation:** 5

**Clarity, Quality, Novelty And Reproducibility:**

Overall, quality and reproducibility are good. While clarity and novelty of the work can be improved.

**Strength And Weaknesses:**

Strength
[+]  Paper is well organized and well motivated. Literature is comprehensive and overall paper is well organized.

Weakness
[-] Novelty seems limited. The algorithm and contribution of the paper is mainly on developing a schema with neural network to handle tabular data and the major source of improvement is from tree based initialization and fine tuning process, which mostly seems to be obtained from Biau et al. (2019).

[-] Some details might need more clarification.
Could the author provides an example or demo graph to illustrate more clearly on what the initialized the neural network look like. Figure 1 is obtained from the original paper to show how tree based model can be converted to neural nets in a intuitive way. Could some demo example listed here to more precisely show the initialization step? Same applies to section 2.3 on the details to convert the model to allow gradient descent training.

[-] Experimental results seem weak.
From the results in table 1, it seems MLP initialized with GBDT still lower perform than GBDT along. Is there any analysis on the source of changes, considering the neural networks is fine tuned on top of GBDT models?


**Summary Of The Paper:**

The paper proposed a new method to initialize neural networks using pertained tree based models for tabular related data. The new proposed framework includes two components:
- [1] As proposed by the prior work 'Neural random forest', 2019, tree based models can be converted into a 3-layer MLP.
- [2] Design the non-linear function with proper selection of parameter to fit into gradient descent training framework.

Numerical result shows that MLP with tree based initialization method converges better than random initialization. Parameters distribution analysis also shows model trained with tree based initialization converges to a solution as compared to random initialization.

**Summary Of The Review:**

The paper considers an interesting and important topics on building a neural network schema to handle tabular data. Initialization with tree based model with designed fine tuning process is reasonable sound. However, the novelty of the paper seems limited since majority of the work seems to be inherited from Biau et al. (2019). Experimental results seem weak as compared to the baseline models. Although there are some interesting findings on the converged models, while there is not theoretical analysis on those behavior.

---

> ### Author Response · Authors · 2022-11-18
> **Specific comments**
>
> **Novelty of the method** Please refer to the general comment for a discussion on the position of our work.
>
>
> **Illustration of the method** We have clarified the principles of the proposed initialization, in particular by providing schematic illustrations in the revised paper, see Figure 7. Figure 7 shows how the initialization step works, namely how from a pre-trained tree-based method, one can initialize the weights of an MLP.
>
>
> **Comparison with GBDT** GBDT is a powerful prediction method with highly optimized implementations (such as XGBoost) and is known to achieve top performances on learning tasks with tabular data. Obtaining a neural network model that is able to compete with GBDT constitutes, in itself, a promising result for NN generalization performances on tabular data. Further, note that the GBDT used for initialization are not the same as the best GBDT used as stand-alone predictors. The best GBDT of Table 1 usually requires a large number of trees with high depth (they are generally composed of 1000 trees of depth 12). This high complexity usually makes a direct translation into a trainable MLP impossible due to width constraints. Therefore, we are forced to use sub-optimal GBDT for the NN initialization. As expected, after training, an MLP with GBDT init. outperforms the light GBDT that was used for its initialisation across all our data sets. The fact that the MLP with GBDT init. reaches performances on par with the ones of the best GBDT is already highly remarkable as the GBDT used for initialisation is much less powerful than the best one. It also shows that the additional layers of the network truly benefit from the tree-based translation of the first two layers. This point has been clarified in the discussion about numerical results.

---

> > ### Comment · Area_Chair_GMyX · 2022-12-03
> > **Reviewer c3Rb: please respond and update your comment**
> >
> > Hi Reviewer c3Rb,
> >
> > Could you please check the authors' response and see if this addresses your concern? The authors provided detail replies but you have not yet replied.
> >
> > This is important before a decision can be reached; if we cannot reach a consensus before the discussion stage concludes, we might need to call a Zoom meeting.
> >
> > Thanks,
> > AC

---

### Author Response · Authors · 2022-11-18
**General comments**


We greatly thank the reviewers for their careful reading, positive comments and insightful suggestions that help to improve the quality of the manuscript. Answers to all points raised in their reviews are discussed below. Modifications in the paper compared to the initially submitted version have been highlighted in green.


**Position of the work** All reviewers discussed the position of our work and in particular its novelty compared to (Biau et al., 2019). In this reference, the authors propose to rewrite a random forest as a neural network architecture composed of 3 layers, with tangent hyperbolic activation functions.  They aim at optimizing tree-based methods using a neural network framework (the neural network inherits the RF structure closely).
In the submitted paper, we adopt a totally opposite perspective by training neural networks with the help of tree-based methods.
We thus consider tree-based methods as relevant feature extractors for tabular data. The partition structure of tree estimators is used to initialize the first layers of an MLP architecture, the other layers being randomly initialized.
Beyond this change of point of view, our approach can be applied to any MLP architecture (of any depth or width), and results in MLP with both dense and sparse layers. Thus, we do not try to mimic exactly the tree behavior, in contrast to (Biau et al., 2019), who consider highly constrained neural networks (composed of exactly 3 layers, all of which have sparse connections). Indeed, we remove the last layer of tree translation at initialization and add extra layers to perform prediction based on the tree-based feature extraction. Besides, new experiments available in the Appendix D.4.2 show that the weights of the two first layers change a lot during training. All in all, our final network has nothing to do with a tree estimator. We have thus modified the paper by clarifying the position of our work with respect to the work of (Biau et al., 2019).



**Additional numerical experiments** We thank the reviewers for pointing out a detailed list of different types of random initialization in neural networks. Note that random (uniform) initialization is the most reliable basis for comparing our new initialization methods, as it is the only method whose performance on tabular data is well known in literature (e.g., Brisov et. al, 2021; Somepalli et. al, 2021) and widely implemented in common packages.
The other initialization methods proposed by the reviewers are developed specifically for CNN. We  have therefore performed new numerical experiments on the most general approaches among them: (i) Xavier initialization (Glorot \& Benigo, 2010) (ii) layer-sequential unit-variance orthogonal initialization (LSUV; Mishkin \& Matas, 2015) and (iii) winning ticket network pruning (Frankle \& Carbin, 2019).
For the first and second methods, we exactly followed the procedure outlined by the authors, using native pytorch functions or code provided by the authors.
Note that the third method is more a simplification approach of the NN architecture during training than an initialization technique. That being said, it remains interesting to compare this strategy to the one developed in the paper, as the winning ticket network pruning enforces the NN sparsity during training. This can be indeed put in parallel to the sparsity of the first layers introduced by the proposed initialization and preserved during training.
The winning ticket network pruning, however, is computationally very intense and has to the best of our knowledge only been studied on medium-sized data sets. We therefore use a slightly different procedure than Frankle \& Carbin to determine winning tickets. First of all, we allocate at most as much computational effort, i.e. training epochs, to determining a winning ticket as we use for the model training itself. This fixed number of training epochs is then distributed among $n$ pruning rounds, each of which consists in training the model, pruning it, and resetting all non-pruned weights to their initial (random) coefficients. This approach takes the same time as one-shot pruning but proves to be more efficient.
Table 1 now includes new results for Xavier, LSUV initialization and the winning ticket approach (due to time constraints, the results are now available for all data sets but Higgs and covertype; this will be completed in the final version). They are all outperformed by the tree-initialized MLP on all but one data sets, which confirms the effectiveness of our method.

---

### Decision · Program_Chairs · 2023-01-20

**Decision:**

Accept: poster

**Justification For Why Not Higher Score:**

The innovations are moderate compared to prior arts; so are the performance improvements.

**Justification For Why Not Lower Score:**

In general, I feel this paper is done solidly, though not particularly exciting. It might be of interest to the specific community

**Metareview: Summary, Strengths And Weaknesses:**

This paper proposed a new sparse initialization technique for (potentially deep) multilayer perceptrons (MLP) towards tabular data. The method uses tree-based techniques to initialize the weights of the NN. The authors argue that using this initialization in the first layer of an MLP is sufficient to improve its performance.

Reviewer c3Rb did not actively engage in the post-rebuttal discussion despite multiple nudges. The other two reviews remain positive. The AC also takes a close look at the paper & comments. The paper is very well written and easy to follow. The proposed method is simple and interesting.

Two main concerns were raised by reviewers:
(1) similarity to prior work  (Biau et al., 2019). The authors clarified that, unlike this prior art, they do not try to mimic exactly the tree behavior,  but instead remove the last layer of tree translation at initialization and add extra layers to perform prediction based on the tree-based feature extractor.
(2) underwhelming performance. The authors added new baseline results with various architecture + initialization ways, and show they are all outperformed by the tree-initialized MLP on all but one data set


**Note From Pc:**

if the above contains the word "oral" or "spotlight" please see: "oral" presentation means -> notable-top-5% and "spotlight" means -> notable-top-25%. As stated in our emails, we are disassociating presentation type from AC recommendations